materials science

molecular dynamics, 1, 1-diamino-2, 2-dinitroethene, polymer-bonded explosives

**Author for correspondence:**
Baoguo Wang
e-mail: wangbaoguo1970@outlook.com

This article has been edited by the Royal Society of Chemistry, including the commissioning, peer review process and editorial aspects up to the point of acceptance.

# Computational analysis the relationships of energy and mechanical properties with sensitivity for FOX-7 based PBXs via MD simulation

Jianbo Fu[1], Baoguo Wang[1], Yafang Chen[1], Yunchuan Li[1], Xing Tan[1], Biyuan Wang[1] and Baoyun Ye[1,2]

[1]North Univ China, Sch Environm and Safety Engn, and [2]North Univ China, Shanxi Engn Technol Res Ctr Ultrafine Powder, Taiyuan 030051, Shanxi, People's Republic of China

(iD) JF, 0000-0002-2190-9809; BaW, 0000-0001-8390-5160

Molecular dynamics (MD) simulations have been applied to investigate 1, 1-diamino-2, 2-dinitroethene (FOX-7) crystal and FOX-7 (011)-based polymer-bonded explosives (PBXs) with four typical polymers, polyethylene glycol (PEG), fluorine-polymer ($F_{2603}$), ethylene-vinyl acetate copolymer (EVA) and ester urethane (ESTANE5703) under COMPASS force field. Binding energy ($E_{bind}$), cohesive energy density (CED), initiation bond length distribution, RDG analysis and isotropic mechanical properties of FOX-7 and its PBXs at different temperatures were reported for the first time, and the relationship between them and sensitivity. Using quantum chemistry, FOX-7 was optimized with the four polymers at the B3LYP/6-311++G(d,p) level, and the structure and RDG of the optimized composite system were analysed. The results indicated that the binding energy presented irregular changes with the increase in temperature. The order of binding ability of different binders to the FOX-7 (011) crystal surface is PEG > ESTANE5703 > EVA > $F_{2603}$. When the temperature increases, the maximum bond length ($L_{max}$) of the induced bond increases and the CED decreases. This result is achieved in agreement with the known experimental fact that the sensitivity of explosives increases with temperature, and they can be used as the criterion to predict the sensitivity of explosives. The descending order of $L_{max}$ is FOX-7 > $F_{2603}$ > ESTANE5703≈EVA > PEG. The intermolecular interactions between FOX-7 and the four polymers were mainly weak hydrogen bonding and van der Waals interactions, and these interactions helped to reduce the bond length of $C-NO_2$, leading to a decrease in the sensitivity of FOX-7. The addition

of polymers can effectively improve the mechanical properties of explosives. Among the four polymers, EVA has the best effect on improving the mechanical properties of FOX-7 (011). At the same temperature, the modulus can be used to predict the sensitivity of high-energy materials. Cauchy pressure can predict the sensitivity of non-brittle energetic materials. The nature of the interaction between FOX-7 and the four polymers is hydrogen bonding and van der Waals force, of which hydrogen bonding is the main one. These studies are meaningful for the formulation design and sensitivity prediction of FOX-7 and its PBXs.

# 1. Introduction

The poor compatibility between high energy and low sensitivity has invariably been a major problem for researchers in the field of energetic materials. 3-nitro-1, 2, 4-triazole-5-one (NTO) [1], 5-amino-3-nitro-1H-1, 2, 4-triazole (ANTA) [2], 1, 1-diamino-2, 2-dinitroethene (FOX-7) [3] and 2, 6-diamino-3, 5-dinitro pyrazine-1-oxide (LLM-105) [4] have been synthesized in order to obtain insensitive high explosives (IHE) with excellent properties. Among them, FOX-7 is one of the most potential candidate varieties and components of IHE. In 1998, Latypov *et al.* first disclosed the synthetic route of FOX-7 [3]. Since its inception, FOX-7 has become the theme of many experimental investigations due to its excellent comprehensive performance. The performance of FOX-7 benefits from its special structure, i.e. 'push-pull' electronic delocalization and inter- and intra-molecular hydrogen bonds [5–7]. Its calculated detonation velocity is 9698 m s$^{-1}$, and crystal density of 1.878 g cm$^{-3}$ is higher that of cyclotrimethylenetrinitramine (RDX, $D = 8800$ m s$^{-1}$, $\rho = 1.82$ g cm$^{-3}$) [8]. Latypov *et al.* used the BAM instrument (2 kg drop hammer) to test and found that the impact sensitivity of FOX-7 was 126 cm, while the impact sensitivity of RDX was 38 cm. They also used the Julius Petri device to test and found that the friction sensitivity of FOX-7 was greater than 350 N, far lower than that of RDX (120 N) [3]. It is verified that FOX-7 has better security than RDX.

Polymer-bonded explosive (PBX) is a kind of mixed explosive with low sensitivity, mechanical strength and high energy density [9–14]. It is mainly composed of explosives, high polymer binders and other components, and has been extensively used in aerospace and military fields. With such excellent performance, the research on optimizing PBXs performance has been widely carried out worldwide [15–20]. Many investigations have demonstrated the superiority of FOX-7-based PBXs [21–23]. Karlsson *et al.* found that PBXs based on FOX-7 could serve as a replacement of Comp B even at rather low solid loadings [21]. Cullis & Townsley found that FOX-7-based PBXs outperformed PBXN-110 [22]. Mishra *et al.* demonstrated that FOX-7 as a replacement for RDX in Comp B showed better thermal stability and lowered shock sensitivity. At the same time, the detonation velocity was comparable with the original formulation. It has the potential to achieve low vulnerability with little sacrifice in performance [23]. FOX-7 has broad application prospects in the field of IHE [7].

Sensitivity is a crucial indicator for evaluating the safety of high-energy materials. Under the stimulation of external environment such as impact, heat, friction, shock wave, etc. the difficulty of the explosion of energetic materials directly affects its synthesis, preparation, transportation, storage and use [24]. Therefore, the research on sensitivity is fundamental and significant. Composite high-energy materials are currently the high-energy materials used in military and aviation applications [25,26]. Hence, the study of sensitivity should not only focus on high-energy components. For high-energy composite materials, attention should be paid to the changes in structure, energy and properties caused by the interaction between high-energy ingredients and other additives, to find the reasons for the thermal decomposition, detonation and sensitivity changes of materials. Molecular dynamics (MD) simulation is a powerful computational tool for complex multi-component systems. In much of the research in the past, individuals have used this method to do a multitude of study on hexanitrohexaazaisowurtzitane (CL-20), RDX, cyclotetramethylenetetramine (HMX), and PBX based on them and calculated the equilibrium structure, binding energy and mechanical properties of different composite systems [27–32]. To explore the microscopic theoretical criteria of the sensitivity of high-energy composite materials, the relations of sensitivity with the cohesive energy density, the bond length distribution and the mechanical properties for the system at different temperatures were also discussed [24,33–37]. At present, there are few studies on the simulation of FOX-7 and its PBXs, while few reports on the research on the relations of sensitivity with structure, energy and mechanical properties for FOX-7 and its PBXs [7]. Although the determination of the sensitivity mainly depends on experiments, it is of considerable significance to save time and reduce experimental risks if the sensitivity can be predicted based on theoretical research.

(a)

H⁺(O—CH₂—CH₂)₁₀OH

(b)

(c)

(d)

**Figure 1.** Chemical structure of PEG (a), EVA (b), F$_{2603}$ (c) and ESTANE5703 (d).

In this paper, to find the microscopic theoretical criteria of the sensitivity of FOX-7 and its PBXs, and the effect of different binders on the performance of FOX-7-based PBXs, four commonly considered polymer binders: polyethylene glycol (PEG, figure 1a), ethylene-vinyl acetate copolymer (EVA, figure 1b), fluorine-polymer (F$_{2603}$, figure 1c) and ester urethane (ESTANE5703, figure 1d) were used, where F$_{2603}$ is copolymers polymerized from vinylidene fluoride and hexafluoropropylene with the molar ratios of 1 : 1. Previous studies have proved that (011) crystal surface of FOX-7 is the vital growth surface and the most paramount crystal face affecting its interaction with the polymer [7]. Therefore, the whole work was carried out with the FOX-7 (011) crystal surface as the base explosive. Four FOX-7 (011)-based PBXs systems were established, and these models were correspondingly abbreviated as FOX-7 (011)/PEG, FOX-7 (011)/EVA, FOX-7 (011)/F$_{2603}$ and FOX-7 (011)/ESTANE5703. They were labelled as P1, P2, P3 and P4, respectively, and the pure FOX-7 (011) system was marked as P. The MD method was used to calculate the mechanical properties, energy and structure of each system. The calculation results were analysed and discussed also in terms of their relations with the sensitivity. The binder with excellent performance was screened out. Energetic materials become more sensitive and less stable, with the temperature increasing [33]. Therefore, the MD simulation is carried out at different temperatures. The mechanical properties and structural data are obtained by calculating from trajectories, and the results are analysed and discussed. The primary purpose of this work is to provide some theoretical guidance for the formulation design of FOX-7-based PBXs and its sensitivity prediction.

# 2. Simulation details

## 2.1. Choice of force field

The reliability of the MD simulation results depends on whether the characteristics of the selected force field are appropriate. Therefore, choosing the right force field is highly valued in the whole work. This paper applies the COMPASS [38] force field for MD simulation. There are two main reasons for choosing this force field. On the one hand, the COMPASS force field can be used to study complex systems, including interfaces and materials, to more accurately simulate the structure and properties of individual or condensed matter. This is because the force field is a powerful all-atomic force field. Most of the parameters are obtained through quantum mechanical calculations combined with empirical optimization. At the same time, it uses a complex set of functions to describe the interaction potential more accurately. On the other hand, the nitro group has explicitly been parametrized and included in the COMPASS force field for excellent suitability for energetic materials [7,39]. Then, the entire work was performed under the COMPASS force field.

## 2.2. Construction of polymer models

For the convenience of comparison with the experiment, the per cent weight of the binders in the PBXs was controlled at about 5% [40]. As shown in figure 1, the chain end groups of PEG, EVA, F$_{2603}$ and

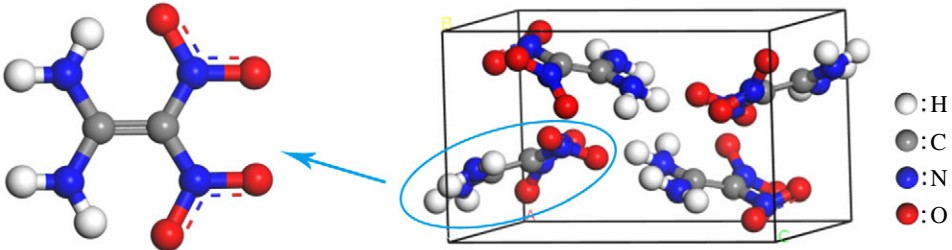

**Figure 2.** Molecular and crystal configurations of FOX-7.

**Table 1.** The details of the four polymer models.

|  | PEG | EVA | F$_{2603}$ | ESTANE5703 |
|---|---|---|---|---|
| the repeating units | $-O-CH_2-CH_2$ | $-CH_2-CH_2-$ $-CH_2-CHOCOCH_3$ | $-CF_2-CF-CF_3-$ $-CF_2-CH_2-$ | $-C_4H_8-COO-$ |
| chain segments | 10 | 12 | 10 | 5 |
| density | 1.27 g cm$^{-3}$ | 0.95 g cm$^{-3}$ | 1.82 g cm$^{-3}$ | 1.25 g cm$^{-3}$ |
| the total number of atoms | 365 | 368 | 154 | 334 |

**Table 2.** Crystal parameters before and after optimization.

| lattice parameter | a (Å) | b (Å) | c (Å) | $\gamma$ (°) |
|---|---|---|---|---|
| experimental | 6.922 | 6.501 | 11.262 | 90.485 |
| computational | 6.799 | 6.710 | 10.814 | 85.625 |
| error % | −1.777 | 3.215 | −3.978 | −5.371 |

ESTANE5703 were saturated by –OH, –CH$_3$ and –H. The details of the four polymer models are shown in table 1. The polymer binder models were optimized for 10 000 steps using the 'Smart' method and placed separately into the amorphous cells. To obtain a relaxed polymer model, the NVT-MD simulation was performed using the Forcite module in Material Studio software, 'Anderson' was selected as the thermostat, the temperature was set at 298 K, the step size was 1 fs, and the total simulation time was 1 ns. Finally, the equilibrium conformation was obtained. The whole process is completed with 'Fine' quality.

## 2.3. Construction of pure FOX-7 (011) and PBX models

The cell structure of the FOX-7 (monoclinic system, P21/N space group, Z = 4) [41] is shown in figure 2. The geometric structure of the single lattice was optimized by the 'Smart' method, and the experimental parameters of the condensed phase were compared with the optimized parameters under the COMPASS force field.

The comparison of experimental lattice parameters with the optimized values of FOX-7 is shown in table 2. The results indicate that the error between the optimized lattice parameters and the experimental values is within 5.5%, which suggests that the COMPASS force field is suitable for simulating FOX-7 [39].

To prevent the polymer from exceeding the boundary of the vacuum layer and avoiding interactions outside the periodic structure, we constructed a 6 × 3 × 4 FOX-7 supercell and cleft the supercell crystal plane along 011. Then a vacuum layer of 10 Å was added along the C direction of the crystal plane to construct a new periodic three-dimensional lattice containing 288 FOX-7 molecules. Taking the P1 system as an example, the entire modelling process is shown in figure 3. The atomic numbers of P, P1, P2, P3 and P4 models are 4032, 4397, 4400, 4186 and 4366, respectively. Next, the 'Smart' method was used again to optimize the configuration of 20 000 steps under the 'Fine' quality. The equilibrium structures of the four binders were placed in the C direction of the optimized FOX-7 (011) crystal

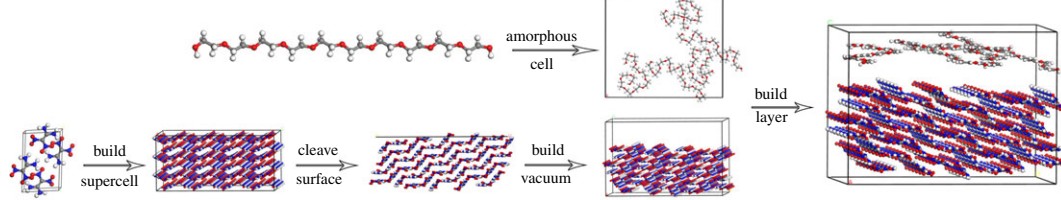

**Figure 3.** The procedure of model establishment of the P1 system.

plane and brought as close as possible to the surface of the FOX-7 molecule. Four different initial models of PBX were obtained.

## 2.4. MD simulations

The simulation process was divided into two parts, one performed in the low-temperature domain (198–398 K) and the other in the high-temperature area (1000–1400 K). The first part of the calculations was performed to study the relationship of energy and mechanical properties with sensitivity for FOX-7-based PBXs. The second part of the calculations was performed at temperatures close to the detonation reaction in order to see how the calculated performance relationship in the low-temperature region changes in the high-temperature region. An explanation of the method can be found in S.4 of the electronic supplementary material.

## 2.5. Simulation at low temperatures

First, the optimized pure FOX-7 (011) and four kinds of PBX models were simulated at temperatures of 198 K, 248 K, 298 K, 348 K and 398 K for 50 ps of NVT-MD. The purpose of this approach is to relax the system further and achieve thermodynamic equilibrium. Then, the PBXs system was gradually compressed by changing the C value. After each compression, 20 000 steps of configuration optimization and 300 ps of NPT-MD simulation were performed until the density reached a stable value. 'Parrinello' was selected as the pressostat, and the pressure is set to 0.0001 GPa, and 'Anderson' was chosen as the thermostat. Electrostatic interactions and van der Waals interactions are calculated by the Ewaled method and the atom-based method. The initial velocity of each molecule was sampled according to the Maxwell–Boltzmann distribution, and the Newton equation of motion was solved by the Verlet method. The van der Waals force was corrected by the cubic spline method, and the cut-off distance was set to 15.5 Å. Throughout the simulation process, the time step was set to 1 fs, and the total number of simulation steps was 300 000 steps. The first 200 000 steps were used for system balance, and the last 100 000 steps were used for statistical analysis. One frame was output per 50 steps, and a total of 2000 structures were saved to analyse static mechanical performance. All these calculations were carried out in Materials studio.

## 2.6. Simulation at high temperatures

Since simulation calculations are more prone to the problem of weak energy conservation at high temperatures, to ensure the reliability of the simulation results, energy conservation test experiments are conducted first. First, select the NVE ensemble at a temperature of 1400 K, using a verlet integrator, the step size is set to 0.1 fs, and the P1 system is simulated for 1 000 000 steps. The calculation results show that the energy drift fluctuates greatly, which cannot guarantee the validity of the simulation results. Change integrator to sixth order symplectic integrator, step size set to 0.1 fs to re-run NVE-MD. The comparison of the results of the two calculations is shown in figure 4. The sixth-order symplectic integrator is selected to recalculate at 0.1 fs steps, and its energy conservation is significantly better than that of the verlet integrator. The next step is to perform NPT-MD of 3 000 000 steps for P, P1, P2, P3 and P4 at temperatures of 1000 K, 1100 K, 1200 K, 1300 K and 1400 K, respectively. 'Parrinello' was selected as the pressostat, and 'Anderson' was chosen as the thermostat, and step size selection 0.1 fs. After the calculation is completed, the equilibrium conformation is collected for result analysis. All these calculations were carried out in Lammps and Materials studio.

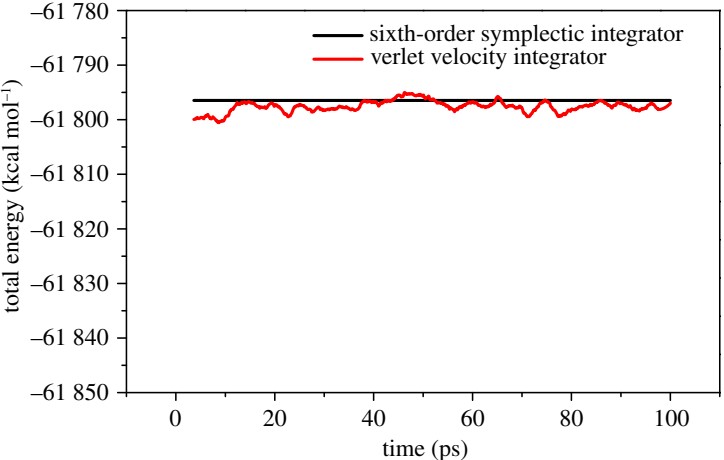

**Figure 4.** Total energy of the 0.1 fs integration step size as a function of time.

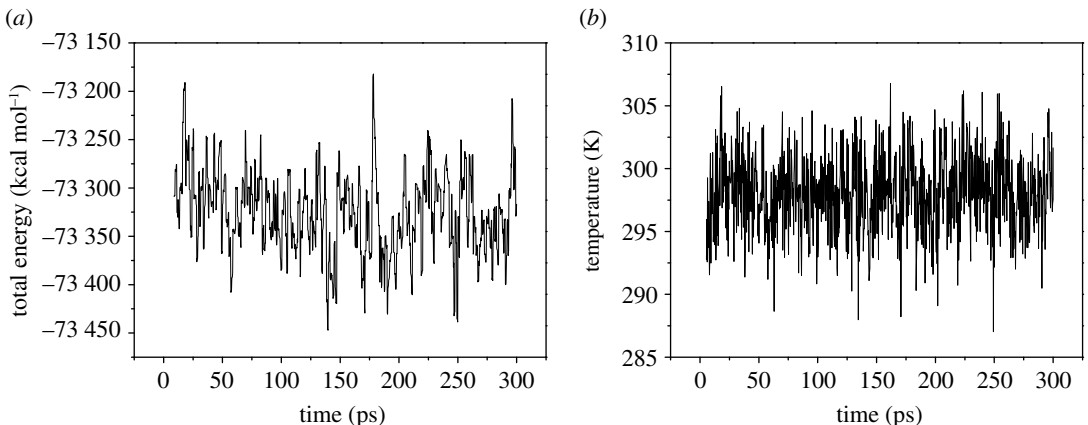

**Figure 5.** Fluctuation curves of energy (*a*) and temperature (*b*) of the PBX with PEG on the molecule layers parallel of FOX-7 (011) crystalline surface.

## 2.7. Quantum chemical calculations

In order to further analyse the structure and interactions between FOX-7 and the four polymeric molecules, four initial structures were designed. The structures are fully optimized at the DFT-B3LYP/ 6-311++G(d,p) level [42]. Local minima were checked without any imaginary frequencies [43]. Plots of RDG versus sign($\lambda_2$)$\rho$ were carried out at the B3LYP/6-311++G(d,p) level. All these calculations were carried out in Gaussian and Multiwfn [44].

# 3. Results and discussion

## 3.1. Judgement of equilibration

System balance is an indispensable prerequisite to calculate the performance of the system accurately. The criterion for the system to reach equilibrium is that the temperature and energy reach equilibrium at the same time. The temperature fluctuations are within the range of ±10 K, and the energy fluctuations are less than 5% [7]. Figure 5 shows the temperature and energy fluctuation curves of the FOX-7 (011)/PEG system at 298 K. As shown in figure 5, the temperature fluctuation range is 20 K, and the energy fluctuation range is 0.2%, which is far less than 5%, so the system has reached a temperature and energy equilibrium state. Figure 6 shows the energy fluctuation of the P1 system after the completion of NVE-MD at high temperatures. The figure shows that there is no apparent drift and discontinuity of energy fluctuations. Figure 7 shows the energy and temperature fluctuations of the P1 system after the completion of NPT-MD at high temperatures. They all reached equilibrium

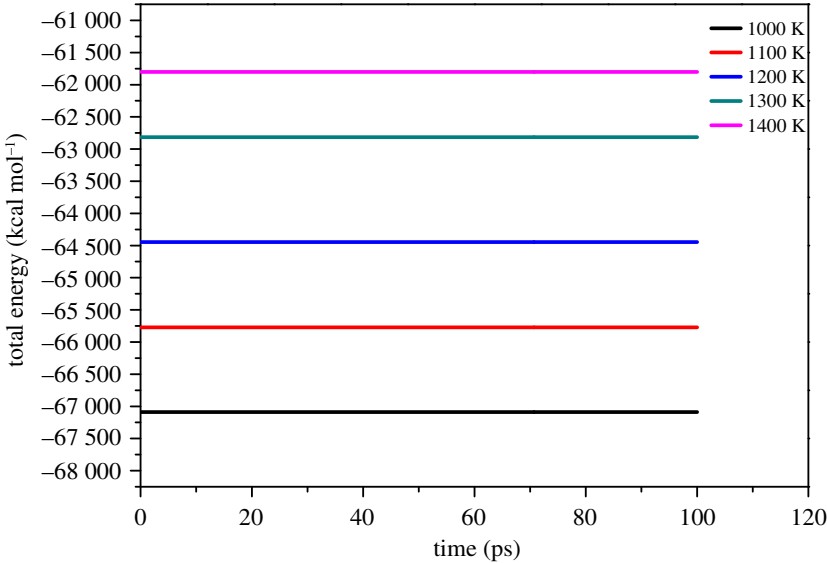

**Figure 6.** Total energy fluctuation of the P1 system after the completion of NVE-MD at high temperatures.

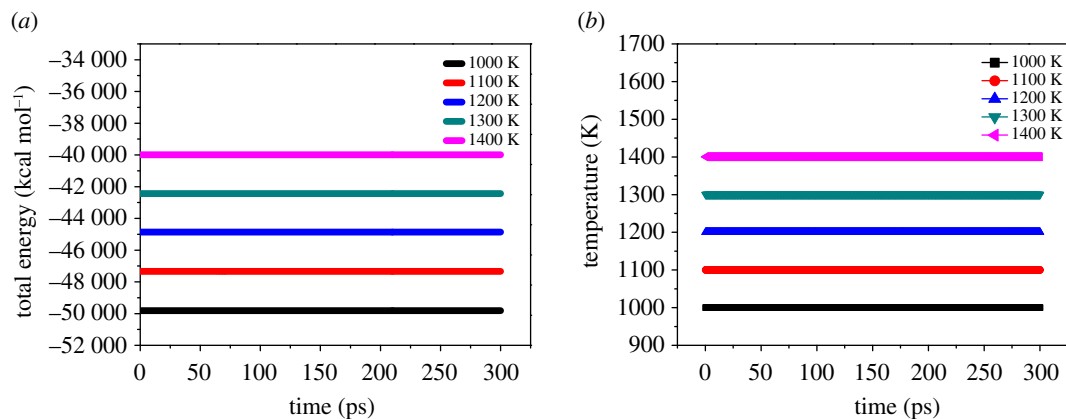

**Figure 7.** Total energy (*a*) and temperature (*b*) fluctuation of the P1 system after the completion of NPT-MD at high temperatures.

on the time scale studied. Other systems have also reached equilibrium and are listed in the electronic supplementary material, figures S1–S8. The equilibrium conformation of the P, P1, P2, P3 and P4 systems is shown in electronic supplementary material, figure S9. All the molecules in the equilibrium conformation are inside the periodic box, which avoids the influence of interactions outside the periodic boundary conditions on the simulation results, provided that an energy balance is established. This proved the rationality of the supercell size.

## 3.2. Binding energy

The compatibility and interaction strength of the two components in the system can be characterized by the binding energy ($E_{bind}$), which directly reflects the ability of FOX-7 crystals to blend with the polymer binder. The intermolecular interaction energy ($E_{inter}$) can be calculated by the total energy of each component of the equilibrium conformation. The $E_{bind}$ is equal to the negative value of the $E_{inter}$. The equation used to calculate the $E_{bind}$ is as follows:

$$E_{bind} = -E_{inter} = -(E_{total} - E_{FOX-7} - E_{polymer}),\quad(3.1)$$

where $E_{total}$ is the total energy of PBX, $E_{FOX-7}$ is the energy of the surface of FOX-7, and $E_{polymer}$ is the energy of the polymer.

Electronic supplementary material, table S1 shows the $E_{bind}$, $E_{total}$, $E_{FOX-7}$ and $E_{polymer}$ of the four binders on the FOX-7 (011) crystal plane at different temperatures. It can be seen from the data in the table that the binding energy of different binders on the FOX-7 (011) crystal plane shows a significant difference. Among the four systems, the $E_{bind}$ of the P1 system at different temperatures was higher

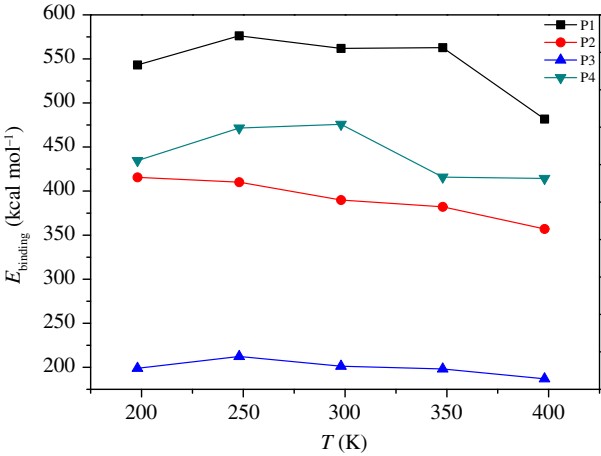

**Figure 8.** $E_{bind}$ versus temperature of P1, P2, P3 and P4.

**Table 3.** The CEDs and CED distribution of P, P1, P2, P3 and P4 at 298 K.

| system | CED (J cm$^{-3}$) | van der Waals (J cm$^{-3}$) | electrostatic (J cm$^{-3}$) |
|---|---|---|---|
| P | $1172 \pm 0.20$ | $538.3 \pm 0.10$ | $619.7 \pm 0.19$ |
| P1 | $1073 \pm 0.18$ | $511.9 \pm 0.09$ | $548.2 \pm 0.17$ |
| P2 | $1042 \pm 0.19$ | $507.7 \pm 0.10$ | $521.3 \pm 0.17$ |
| P3 | $1067 \pm 0.19$ | $498.4 \pm 0.10$ | $555.0 \pm 0.18$ |
| P4 | $1057 \pm 0.19$ | $515.7 \pm 0.09$ | $527.4 \pm 0.17$ |

than the other systems, and the advantages were evident, indicating that PEG was a better FOX-7 binder. The binding energy of the P3 system was the lowest, and P4 was slightly better than P2. As shown in figure 8, the $E_{bind}$ of the four PBXs varies irregularly with increasing temperature. The $E_{bind}$ of P1 and P4 showed an intricate change pattern of rising first, then falling and then rising with increasing temperature. The $E_{bind}$ of P3 between 198 and 248 K increases with temperature and then decreases with further temperature rise. The $E_{bind}$ of P2 regularly decreases with increasing temperature. This complex change is understandable because of the many factors that affect the binding energy, and only characterizes the overall thermal stability of the system. Since the sensitivity acts within the 'hot spot' theory as a local property, $E_{bind}$ cannot be used to predict sensitivity [24].

## 3.3. Cohesive energy density

Cohesive energy density (CED) refers to the energy required for 1 mol aggregates per unit volume to overcome intermolecular forces to become gaseous. This is a physical quantity that evaluates the magnitude of intermolecular forces and can be measured experimentally. In general, the stronger the intermolecular force, the higher the CED value [45–47]. In MD simulation, van der Waals force and electrostatic force determine the value of CED, and the hydrogen bonding between polar groups also affects CED to a certain extent. The larger the CED value, the higher the energy required for the substance to change from the condensed phase to the gas phase, and the more difficult it is for decomposition and explosion to occur. Table 3 lists the values of CED, van der Waals and electrostatic forces of P, P1, P2, P3 and P4 systems at 298 K. Figure 9 shows the CED of five systems as a function of temperature.

In table 3, it can be found that the CED values of P1, P2, P3 and P4 are all smaller than P. This phenomenon may be due to the high regularity of pure FOX-7 (011) crystals. This well-defined structure makes it more difficult to remove a single molecule. The addition of the binder affects the original regularity of the FOX-7 crystal, making the CED value of the PBXs system lower than P. Although the CED of PBXs is reduced, it does not mean that the addition of the binder makes the system more sensitive. The passivation effect caused by the thermal insulation and heat absorption effect of polymer spread on the surface of explosives exceeds the sensitization effect caused by CED reduction. That is why system P shows the highest CED value, but the sensitivity is still higher than

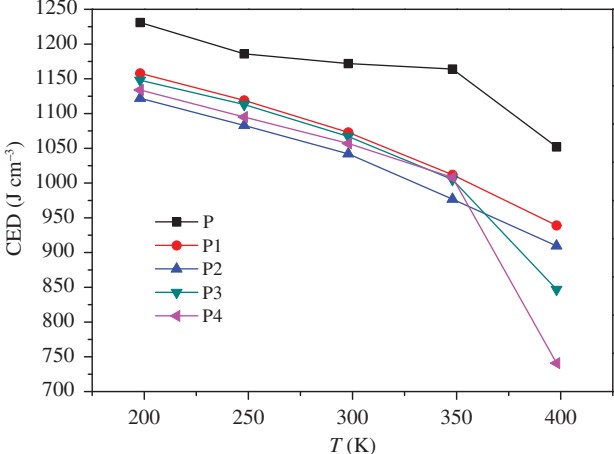

**Figure 9.** CEDs versus temperature of P, P1, P2, P3 and P4.

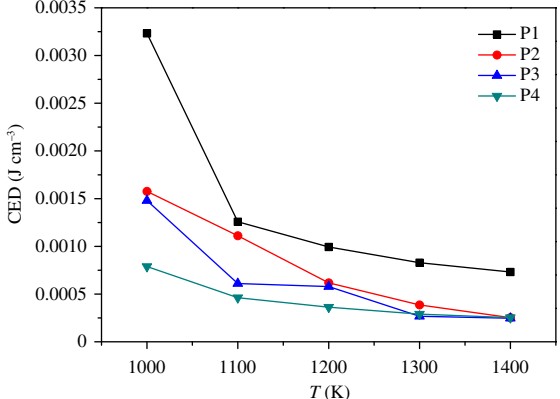

**Figure 10.** CEDs versus temperature of P, P1, P2, P3 and P4.

PBXs. Therefore, comparing the CED value between the same types of systems can better correlate the sensitivity level.

It can be observed from figure 9 that the CED values of the five systems gradually decrease with increasing temperature, indicating that the energy required for the evaporation of the system to overcome the intermolecular force decreases. This phenomenon is achieved in agreement with the known experimental result that the sensitivity increases with increasing temperature. It suggests that under the condition of increasing temperature, CED can be used to judge the relative magnitude of sensitivity. After careful consideration of the CEDs of four PBXs at different temperatures, the ranking is P1 > P2 > P3 > P4.

At high temperatures, the energy required for the material to transfer from the condensed phase to the gas phase becomes extremely low, which easily leads to decomposition and explosion, resulting in higher sensitivity. As shown in figure 10, the CED of PBXs at high temperatures is significantly reduced compared with the CED at low temperatures. It can be found from figure 10 that the CED values of the four PBXs decrease with increasing temperature, which is consistent with the results obtained at low temperatures. CED can be used as a theoretical criterion for thermal sensitivity and impact sensitivity. At 1000 K, there is a certain gap between the CED values of the four groups of systems, but they are all in the same order of magnitude, and the CED value of P1 is the largest. When the temperature rises to 1400 K, the CED values of the P2, P3 and P4 systems approach the same, while the P1 system still maintains a certain degree of insensitivity. According to the changes of CED values of the four PBXs under the temperature of 1000–1400 K, the ranking is P1 > P2 > P3 > P4.

## 3.4. Initiation bond length distribution

As we all know, the bond order represents the difficulty of chemical bond breaking. In chemical reactions, the smaller the order of a chemical bond, the longer its length, and the easier it is to be broken. In

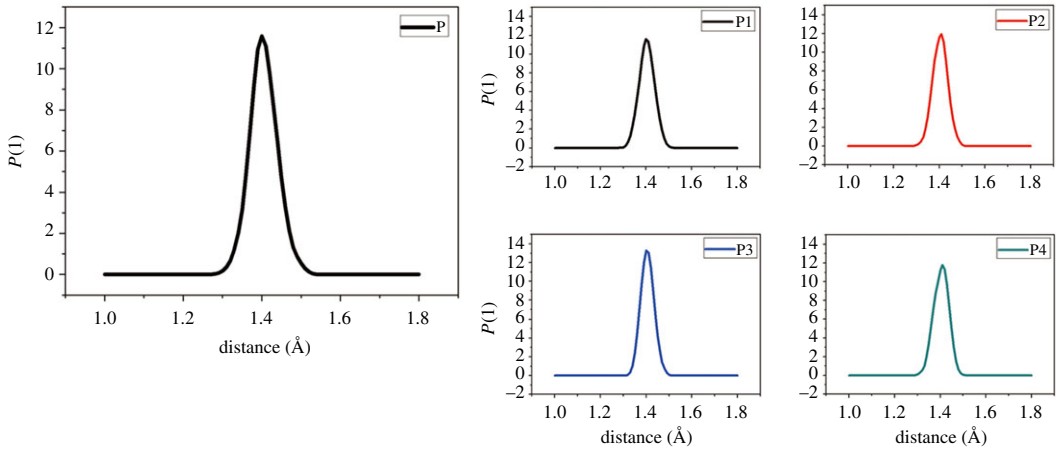

**Figure 11.** Initiation bond length distributions of P, P1, P2, P3 and P4 (at 298 K).

**Table 4.** $L_{max}$ and $L_{ave}$ (Å) of initiation bond length in P, P1, P2, P3 and P4 (at 298 K).

|  | P | P1 | P2 | P3 | P4 |
|---|---|---|---|---|---|
| $L_{max}$ | 1.5068 | 1.4902 | 1.4951 | 1.4835 | 1.4839 |
| $L_{ave}$ | 1.4057 | 1.4036 | 1.4047 | 1.4053 | 1.4042 |

energetic materials, the initiation bond is the most unstable chemical bond, and it is easily triggered and exploded when stimulated by external conditions. Therefore, the research on the sensitivity of high-energy materials should focus on its initiation bond length distribution. Previous *ab initio* MD simulations and molecular orbital calculations have inferred that there are multiple ways of thermal decomposition of FOX-7, among which C-NO$_2$ bond cleavage and FOX-7 isomerization are the main starting decomposition methods [48,49]. In the face of large and complex PBX systems, general quantum chemical methods are not applicable. Although MD simulation does not involve electronic structures and cannot provide bond order data between atoms, it can give bond length distributions of equilibrium structures at different temperatures, to obtain essential laws that can be related to the actual situation. Table 4 lists the maximum bond length ($L_{max}$) and average bond length ($L_{ave}$) of the FOX-7 (011) and its PBXs at 298 K. Figure 11 shows the initiation bond length distribution of each system of P, P1, P2, P3 and P4 at 298 K.

From the data in table 4, it can be concluded that the addition of four polymer binders causes the values of $L_{max}$ and $L_{ave}$ to decrease slightly. The bond lengths of the five systems in figure 11 showed symmetric Gauss distribution. The most probable bond lengths ($L_{prob}$) of the four PBX systems are almost unchanged compared with the P system, but the distribution width is slightly increased compared with the P system. It is because of the addition of the binder that the initiation bond in FOX-7 (011) is more orderly and stable. Therefore, it is not difficult to find that the formation of the PBX system is beneficial to reduce the sensitivity of explosives.

$L_{max}$ is a more critical parameter than $L_{ave}$. Although the ratio is lower, it is overwhelmingly active. Molecules with $L_{max}$ often undergo the first chemical reaction, which in turn triggers the decomposition and explosion of explosives. From figure 12a,b, we can find that both $L_{ave}$ and $L_{max}$ increase with increasing temperature. This phenomenon is achieved in agreement with the experimental result that the sensitivity of energetic materials becomes more sensitive to increasing temperature. From figure 12, we can also find that the $L_{max}$ of P is higher than the PBXs system, and the $L_{ave}$ value of P is the highest in the temperature range of 298 to 398 K, indicating that the sensitivity of PBXs is lower than that of pure FOX-7 (011). This is mainly due to the excellent ductility of the polymer, which plays a role in absorbing shock and heat when the explosive is stimulated by external energy. Pure FOX-7 (011) has the highest sensitivity, and the descending order of the initiation bond length of the four PBXs is P3 > P4 ≈ P2 > P1.

As shown in figure 13, the length change of $L_{max}$ at high temperature is pronounced compared with that at low temperatures. As the temperature increases, $L_{max}$ and $L_{ave}$ gradually become more substantial, but the P1 system is always lower than other systems. At 1400 K, the $L_{max}$ of P, P2, P3 and P4 are almost the same, which indicates that their sensitivity is extremely high at this time; they are

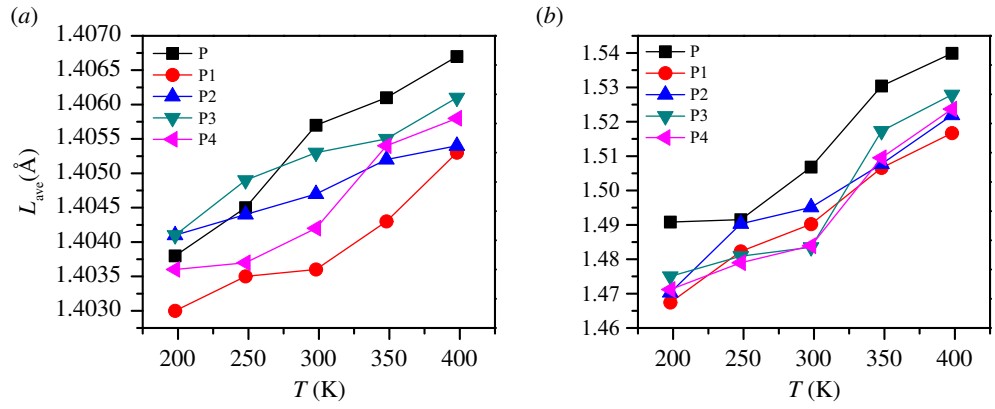

**Figure 12.** $L_{ave}$ (a) and $L_{max}$ (b) of initiation bond length versus temperature of P, P1, P2, P3 and P4.

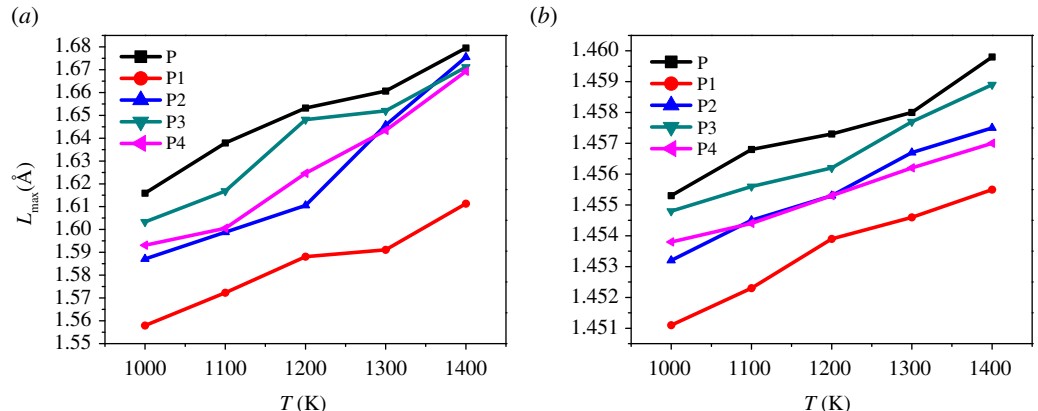

**Figure 13.** $L_{max}$ (a) and $L_{ave}$ (b) of initiation bond length versus temperature of P, P1, P2, P3 and P4.

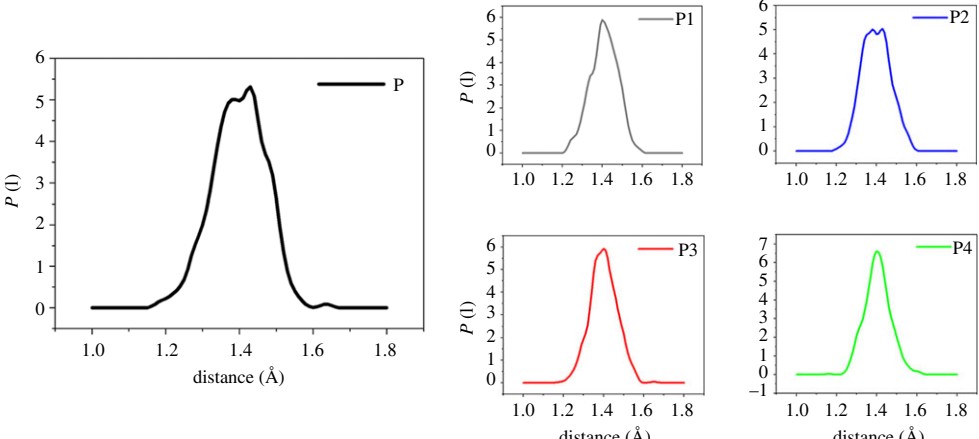

**Figure 14.** Initiation bond length distributions of P, P1, P2, P3 and P4 (at 1400 K).

already at the critical point of explosion or have been exploded, but the P1 system still maintains a certain degree of insensitivity, which phenomenon is consistent with the rule presented by CED at high temperatures. It can be seen from figure 14 that $P(l)$ in the high temperature domain is significantly lower than that in the low temperature domain, and the curve no longer exhibits a symmetric Gaussian distribution. This is mainly due to the elongation or breakage of chemical bonds at high temperatures and the intense movement of molecules. $L_{max}$ is most easily activated at high temperatures, so $L_{max}$ can directly reflect the impact sensitivity of PBX, and its impact sensitivity descends as follows: P1 < P2 ≈ P4 < P3, which is consistent with the results calculated at low temperatures.

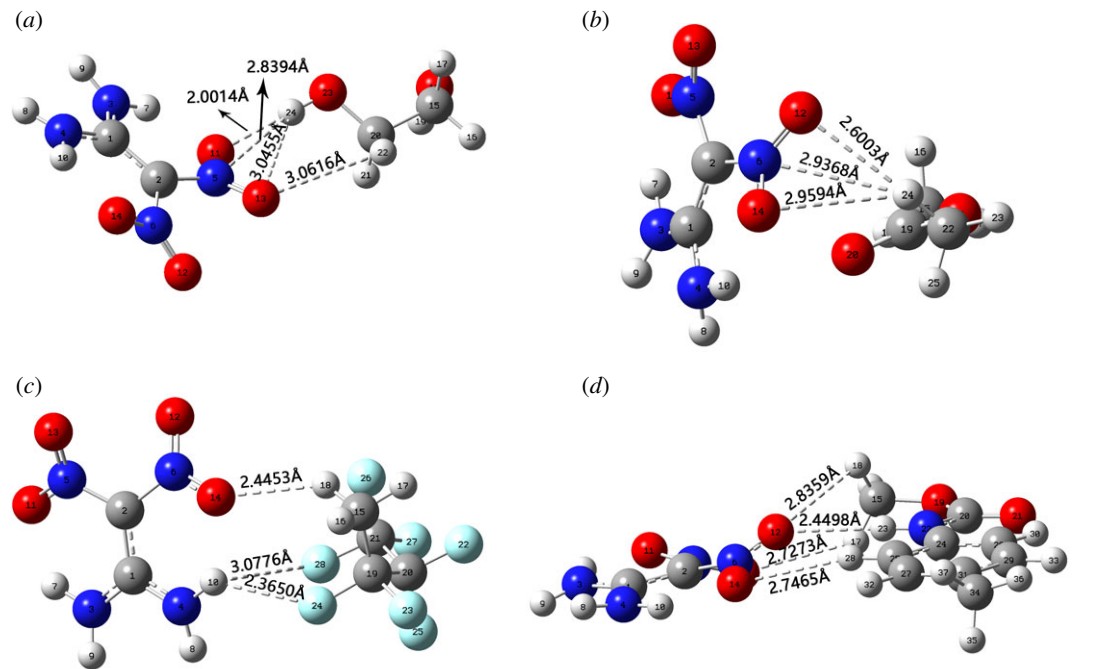

**Figure 15.** Optimized structures and atomic numbers of P1 (*a*), P2 (*b*), P3 (*c*) and P4 (*d*) at B3LYP/6-311++G(d,p) level. Dashed lines represent H-bonds. Grey, white, red, blue and cyan stand for C, H, O, N and F atoms, respectively.

**Table 5.** Bond length (in Å) of FOX-7 in monomer and PBXs.

| parameters | monomer | | P1 | P2 | P3 | P4 |
| --- | --- | --- | --- | --- | --- | --- |
| | EXP | FOX -7 | | | | |
| C1-C2 | 1.446[a] | 1.422 | 1.427 | 1.422 | 1.426 | 1.428 |
| C2-N5 | 1.419[a] | 1.436 | 1.425 | 1.433 | 1.435 | 1.432 |
| C2-N6 | 1.419[a] | 1.436 | 1.430 | 1.432 | 1.433 | 1.430 |
| N5-O11 | 1.237[a] | 1.250 | 1.258 | 1.251 | 1.249 | 1.247 |
| N5-O13 | 1.231[a] | 1.217 | 1.217 | 1.219 | 1.218 | 1.218 |
| N3-H9 | 0.823[a] | 1.007 | 1.007 | 1.007 | 1.007 | 1.007 |
| N4-H8 | 0.856[a] | 1.007 | 1.007 | 1.009 | 1.008 | 1.007 |
| N4-H10 | 0.913[a] | 1.015 | 1.015 | 1.015 | 1.014 | 1.015 |

[a]Denote experiment data derived from [50].

## 3.5. Structure analysis

Figure 15 shows the assembly structure of the four PBXs. All four structures were optimized at the B3LYP/6-311++G(d,p) level, and the optimization results are listed in table 5. The optimized values are close to the experimental values, indicating the applicability of B3LYP/6-311++G(d,p) to the FOX-7 system. In combination with the data in figure 15 and table 5, it is easy to see that the C2-N5 and C2-N6 bonds in the PBXs system are shorter than those in the monomer. When a polymeric molecule forms a hydrogen bond with an O or N atom of $C\text{-}NO_2$ in FOX-7, the bond length of $C\text{-}NO_2$ becomes shorter, and the degree of shortening increases with the strength of the hydrogen bond. It can be found that the non-bonding interaction between the polymer molecule and the FOX-7 molecule may be one of the reasons why the polymer is able to reduce the sensitivity to explosive. Table 6 shows in detail the hydrogen bonding parameters of FOX-7 with four polymer molecules, distances of H ··· A for O(23)H(24) ··· O(11) (2.0014 Å), O(23)H(24) ··· N(5) (2.8394 Å), C(20)H(22) ··· O(13) (3.0616 Å) and O(23)H(24) ··· O(13) (3.0455 Å) in P1, C(22)H(24) ··· O(12) (2.6003 Å), C(22)H(24) ··· O(14) (2.9594 Å) and C(22)H(24) ··· N(6) (2.9368 Å) in P2, N(4)H(10) ··· F(24) (2.3650 Å), N(4)H(10) ··· F(28) (3.0776 Å) and C(15)H(18) ··· O(14) (2.4453 Å) in P3, C(15)H(17) ··· O(13) (2.7273 Å), C(15)H(18) ··· O(12) (2.8359 Å),

**Table 6.** Hydrogen bonding distance (in Å) and angle (in degree) for the H⋯A interactions of P1, P2, P3 and P4.

| PBXs | donor-H⋯accepter | D-H | H⋯A | D⋯A | D-H⋯A |
|------|------------------|-----|-----|-----|-------|
| P1 | O(23)H(24)⋯O(11) | 0.9666 | 2.0014 | 2.0680 | 179.9949 |
|    | O(23)H(24)⋯N(5) | 0.9666 | 2.8394 | 3.7497 | 157.2985 |
|    | O(23)H(24)⋯O(13) | 0.9666 | 3.0455 | 3.7889 | 134.7648 |
|    | C(20)H(22)⋯O(13) | 1.0995 | 3.0616 | 3.8612 | 129.9892 |
| P2 | C(22)H(24)O(12) | 1.0899 | 2.6003 | 3.6260 | 156.5220 |
|    | C(22)H(24)O(14) | 1.0899 | 2.9594 | 3.8104 | 135.1450 |
|    | C(22)H(24)N(6) | 1.0899 | 2.9368 | 3.8678 | 143.5247 |
| P3 | N(4)H(10)F(24) | 1.0138 | 2.3650 | 3.1304 | 131.5177 |
|    | N(4)H(10)F(28) | 1.0138 | 3.0776 | 3.6490 | 116.8231 |
|    | C(15)H(18)O(14) | 1.0888 | 2.4453 | 3.3122 | 135.6174 |
| P4 | C(15)H(17)O(13) | 1.0925 | 2.7273 | 3.5230 | 129.3613 |
|    | C(15)H(18)O(12) | 1.0937 | 2.8369 | 3.4470 | 115.2213 |
|    | N(22)H(23)O(12) | 1.0096 | 2.4498 | 3.4496 | 170.4734 |
|    | C(25)H(28)O(14) | 1.0845 | 2.7465 | 3.8019 | 164.3198 |

N(22)H(23) ⋯ O(12) (2.4498 Å) and C(25)H(28) ⋯ O(14) (2.7465 Å) in P4 belong to the range hydrogen bonding interactions. The hydrogen bonding range from 1.10 to 3.10 Å [51]. The shorter the hydrogen bonding distance and the closer the bond angle is to 180°, the stronger the hydrogen bond [42]. From this, we can conclude that the hydrogen bonding strengths in the four systems have the following descending order: P1 > P4 > P3 > P2.

## 3.6. RDG analysis

Reduced density gradient (RDG) can reveal non-covalent bonding interactions in real space based on electron density and can be used to characterize weak intermolecular interactions [43]. Figure 16 shows the scatter plot and isosurface plot of RDG for the composite structures of FOX-7 molecules with four polymer molecules. The vertical coordinate of the scatter plot represents the RDG and the horizontal coordinate represents the electron density multiplied by the sign of the second Hessian eigenvalue (sign($\lambda_2$)$\rho$). In figure 16, the low-density value surfaces represent weak non-bonding interactions (van der Waals (VDW) forces), while the high-density value surfaces represent strong non-bonding interactions (attractive H-bond and steric clashes, respectively) [42]. From the colour distribution of the isosurfaces in figure 16, hydrogen bonding and van der Waals interactions play a major role in intermolecular interactions. Thus the RDG analysis is expanded in the range of −0.04 to 0 arb. units of the sign($\lambda_2$)$\rho$ value.

Some differences in the scatter distribution of the four systems in the range of −0.04 to 0 arb. units can be seen in figure 16, and the presence of peaks can also be observed. This suggests that non-bonding interactions exist in every system and that there is some variation in the distribution and strength of such interactions. Comparing the RDG plots of P1, P2, P3 and P4, it is easy to see that the first peak occurs between −0.04 and −0.03 arb. units, and the blue colour is the darkest. The first peak represents strong hydrogen bonding within the FOX-7 molecule, mainly between the H atom in the amine and the O atom in the nitro. The scatter between −0.03 and −0.01 arb. units is blue-green, and peaks in this range represent weak hydrogen bonding. The scatter distribution in this interval shows that the P1 and P4 systems have two peaks in this interval and a greater number of scattered points, which indicates that the hydrogen bonding is stronger in the P1 and P4 systems than in the P2 and P3 systems. In the range from −0.01 to 0 arb. units, the colour turns completely green, and the scatter in this region represents the van der Waals force. The van der Waals forces in P1, P2, P3 and P4 are all present with comparable strengths, but the green scatter in P4 is smaller and the van der Waals strength is weaker than in the other systems, as can be seen in figures 16*d*. For the four systems considered, weak hydrogen bonding and van der Waals force interactions are the main interactions between FOX-7 molecules and polymer molecules.

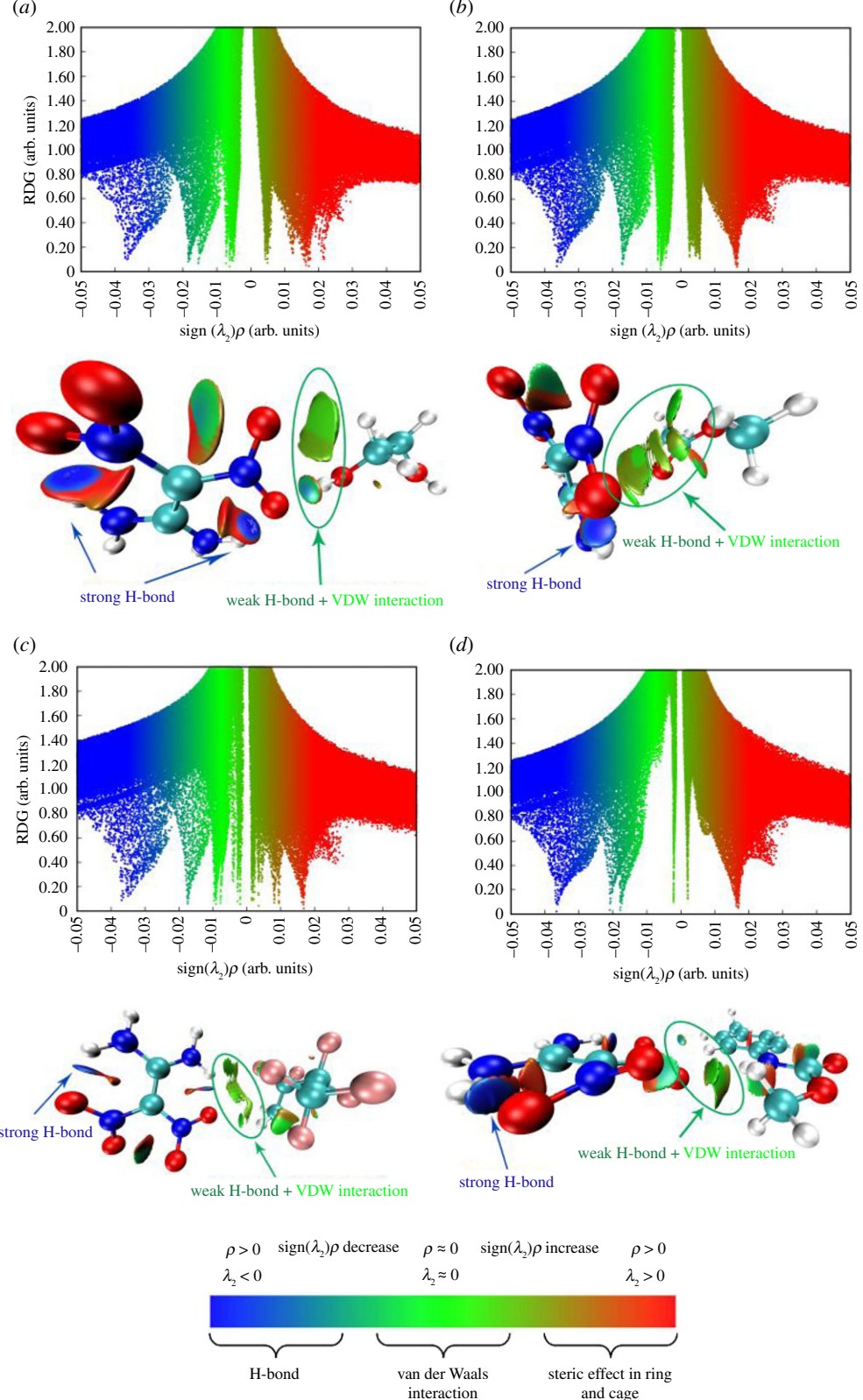

**Figure 16.** (*a*–*d*) The scatter diagram and isosurface graph of RDG for P1, P2, P3 and P4 composite structures.

## 3.7. Mechanical properties

The excellent mechanical properties of energetic materials are a prerequisite for their ability to be processed and produced, and a crucial factor in determining their safety. According to elastic mechanics, stress and strain can reflect the inherent properties of materials, and they all obey Hooke's

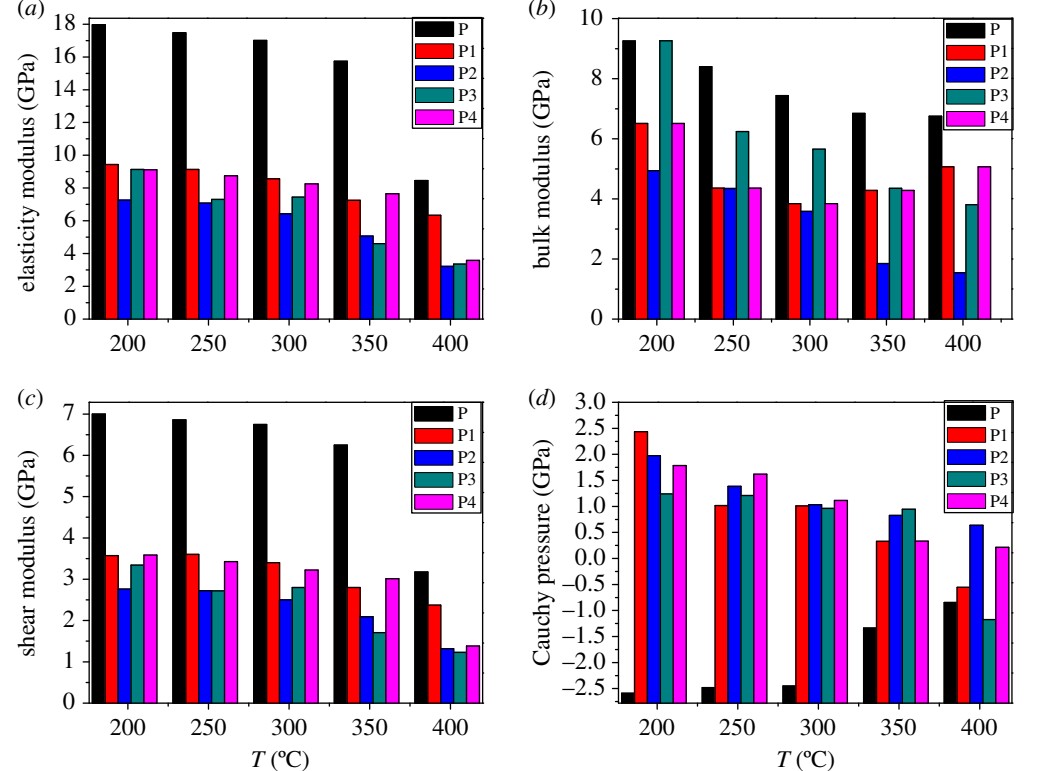

**Figure 17.** (a–d) Mechanical properties versus temperature of P, P1, P2, P3 and P4.

**Table 7.** Mechanical properties of FOX-7 and FOX-7-based PBXs at 298 K.

| SYSTEM | elasticity modulus (GPa) | bulk modulus (GPa) | shear modulus (GPa) | Cauchy pressure (GPa) | Poisson's ratio | K/G |
|---|---|---|---|---|---|---|
| P | 17.00 | 7.44 | 6.74 | −2.45 | 0.26 | 1.10 |
| P1 | 8.56 | 3.84 | 3.40 | 1.01 | 0.26 | 1.13 |
| P2 | 6.43 | 3.59 | 2.50 | 1.03 | 0.29 | 1.43 |
| P3 | 7.45 | 5.65 | 2.80 | 0.96 | 0.33 | 2.02 |
| P4 | 8.25 | 4.35 | 3.22 | 1.11 | 0.28 | 1.35 |

Law [52]. Fundamental theory of mechanical performance is available in electronic supplementary material, S.3.1.

Table 7 summarizes the isotropic mechanical properties of FOX-7 (011) and FOX-7 (011)-based PBXs at 298 K. The change in mechanical properties with temperature is shown in figure 17. From the data in table 7 and figure 17, the addition of the binder caused the E, K and G of FOX-7 (011) to decrease significantly. In terms of mouldability, the Cauchy pressure of P is negative, indicating that the material is brittle. This situation is improved with the addition of the binder. The Cauchy pressures of P1, P2, P3 and P4 are all positive values, which indicates that the addition of the polymer increases the ductility of the system based on Cauchy pressure, and the material shows toughness. Similarly, the K/G values of the four PBXs systems are greater than P, indicating that the addition of polymers increases the K/G-based ductility of the system. The above shows that the formation of the PBX system effectively reduces rigidity, enhances elasticity and formability, and is achieved in agreement with the known experimental result that the addition of a binder to the explosive system can reduce sensitivity. As the preparation and use of PBXs are mostly carried out at room temperature, it is more practical to discuss the mechanical properties of materials at 298 K. At 298 K, the descending order of E, K and G are: P > P1 > P4 > P3 > P2, P > P3 > P4 > P1 > P2, P > P1 > P4 > P3 > P2. It can be seen that EVA is the best choice to improve the overall mechanical properties of FOX-7-based PBX at room

**Table 8.** Tensile strength of FOX-7 columns.

| no. | solvent | solute | experimental E (GPa) | computational E (GPa) |
|-----|---------|--------|----------------------|-----------------------|
| P1 | ethyl acetate | PEG | 5.83 | 8.56 |
| P2 | ethyl acetate | EVA | 4.97 | 6.43 |
| P3 | ethyl acetate | $F_{2603}$ | 5.25 | 7.45 |
| P4 | ethyl acetate | Estane5703 | 5.71 | 8.25 |

temperature. To verify the effectiveness of the mechanical performance simulation, four groups of PBX samples P1, P2, P3 and P4 were selected to perform tensile strength experiments at 298 K. The experimental process can be viewed in the electronic supplementary material, S.3.2. The experimental results are shown in table 8. The results show that there is a certain deviation between the tensile strength of the coated FOX-7 sample and the MD simulation results. The experimental results are smaller than the simulation results, but they are within the same level of a reasonably acceptable range. From the data in table 8, we can see that the descending order of the E values of the four groups of PBXs obtained from the simulation results is: P1 > P4 > P3 > P2, which is consistent with the experimental results. Therefore, the simulation method and the established model are effective for the screening of PBX formula, and the establishment of the model is also accurate.

Besides, we can further see from figure 17 that the mechanical property parameters change with temperature. The values of E, K and G in each system decreased with increasing temperature. Such a change is understandable. As the temperature increases, the kinetic energy of the FOX-7 molecule increases, and at the same time, the conformational change of the segment of the polymer is accelerated, and the flexibility is enhanced, thereby making the material easier to deform. Therefore, as the temperature increases, the rigidity of the system decreases and softens, which is in line with the experimental fact that the mechanical properties are gradually changed. Still, it does not match the empirical fact that the sensitivity increases with temperature. This phenomenon occurs because the sensitization effect caused by the elongation of the initiation bond length of the system as the temperature rises far exceeds the passivation effect caused by the decrease in modulus. Therefore, the modulus is applicable for correlating the sensitivity of different systems at the same temperature. Figure 17*d* is a graph of the Cauchy pressure versus temperature for each system. The Cauchy pressures of P1, P2, P3 and P4 all decrease gradually with increasing temperature, indicating that their ductility based on Cauchy pressure decreases. This trend is achieved in agreement with the known experimental result that the sensitivity of PBXs increases with temperature. This shows that the value of Cauchy pressure can correlate the relative magnitude of PBX systems sensitivity. Interestingly, the Cauchy pressure of the P system is invariably less than 0 at different temperatures. However, as the temperature rises, the Cauchy pressure increases, indicating that the brittleness of P gradually weakens, which is caused by the increase in kinetic energy of the FOX-7 molecule as the temperature rises. This phenomenon also proves that for brittle pure component high-energy materials, Cauchy pressure cannot be used to correlate its sensitivity. It can be seen that Cauchy pressure can only be related to the relative magnitude of the sensitivity of non-brittle high-energy materials.

## 4. Conclusion

In this paper, MD simulations of FOX-7 (011) and FOX-7 (011)-based PBXs were performed. FOX-7 was optimized with four polymers at the B3LYP/6-311++G(d,p) level using quantum chemical methods. The significant findings can be summarized as follows:

(1) The binding energy can evaluate the overall thermal stability of the system, but it cannot be used to predict sensitivity. The descending order of the binding energy of the four PBXs systems is P1 > P4 > P2 > P3.

(2) The CED values decreases with the increase of temperature. At different temperatures, CED can be used as a theoretical judgement of sensitivity. The CED values are ranked as P1 > P2 > P3 > P4.

(3) The calculations at both high and low temperatures indicate that $L_{\max}$ is an effective structural parameter that can be used to predict impact susceptibility. The order of the impact sensitivity of the four FOX-7-based PBXs, as determined by the value of $L_{\max}$, is P1 < P2 ≈ P4 < P3.

(4)  The addition of polymers can effectively improve the mechanical properties of the PBX system. The modulus of each system decreases with the increase in temperature and has a negative correlation with the increasing trend of sensitivity. Cauchy pressure can be used as a criterion for the sensitivity of non-brittle energetic materials.

(5)  The RDG reveals the interfacial interaction between FOX-7 and the polymer in the form of mainly weak hydrogen bonding and van der Waals forces.

In conclusion, the entire study shows that PEG is more conducive to the reduction of sensitivity of the FOX-7-based PBX system, and EVA can make FOX-7-based PBX system have better mechanical properties. This work provides theoretical criteria for the sensitivity of FOX-7 and its PBXs, and also provides a basis for the selection of binders.

Ethics. This article does not present research with ethical considerations.

Data accessibility. Our data are deposited at the Dryad Digital Repository: https://doi.org/10.5061/dryad.bcc2fqz8f [53].

Authors' contributions. Ba.W. mainly provided the article's conception and feasibility analysis; J.F. completed the whole simulation work and wrote the manuscript; Y.C., Y.L., X.T. and Bi.W. put forward suggestions and helped revise the manuscript, and B.Y. provided help with the software.

Competing interests. We declare we have no competing interest.

Funding. Financial support came from the National Natural Science Foundation of China (grant no. 11602231) and Fundamental Science on Underground Target Damage Technology Laboratory Project (grant no. DXMBJJ2017-09).

Acknowledgement. We gratefully thank the National Natural Science Foundation of China (no. 11602231) and Fundamental Science on Underground Target Damage Technology Laboratory Project (no. DXMBJJ2017-09) for their support.

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
