## [Peer Review File · Royal Society Open Science]

Review History

RSOS-200345.R0 (Original submission)

Review form: Reviewer 1

Is the manuscript scientifically sound in its present form?

No

Are the interpretations and conclusions justified by the results?

Yes

Is the language acceptable?

No

Do you have any ethical concerns with this paper?

No

Have you any concerns about statistical analyses in this paper?

No

Recommendation?

Major revision is needed (please make suggestions in comments)

Comments to the Author(s)

“shock wave sensitivity” should “shock sensitivity”; CL-20, RDX, HMX should be explained with full chemical name first; Codes A,B,C,D are not mentioned in caption of Figure 1; Captions of various figures such as Fig.8 is not below the figure; “This is since” should be “this is because”;

there are much more clarity and grammar problems in the text;

Abstract and conclusion sections should be largely briefed with only key information;

When citing the literature, mentioning the family name is sufficient, e.g. “Vijaya S. MISHRA” could be briefed as “Vijaya”;

In Table 1, the experimental lattice parameters should be provided for FOX-7;

There is no relevant results presented on sensitivity of FOX-7 based PBXs, so the title of this paper is misleading;

Over the temperature of 300 °C, the FOX-7 should undergo decomposition, and is it reasonable to investigate the mechanical properties under such high temperature?

At least one group of mechanical parameters should be compared with experimental ones, so that the models build in this paper could be validated;

The systems of P, P1 (only roughly shown in Figure 3) to P4 are not clear in structures, without providing any information on the assembling of the molecules, the repeating units of the polymers, the density of the packing, the total number of atoms for simulations;

The theoretical background of mechanical properties are well-known in the literature (equation 2 to 7 with relevant text), and should be moved to supporting information; there are some less important figures and tables could also be included in supporting information for overall simplicity and clarity; Similar results as Figure 4 for the other mixture systems should be provided in supporting information as well;

Review form: Reviewer 2

Is the manuscript scientifically sound in its present form?

No

Are the interpretations and conclusions justified by the results?

No

Is the language acceptable?

Yes

Do you have any ethical concerns with this paper?

No

Have you any concerns about statistical analyses in this paper?

No

Recommendation?

Major revision is needed (please make suggestions in comments)

Comments to the Author(s)

Journal: Royal Society Open Science

Manuscript ID RSOS-200345

Title: Computational analysis of energy, mechanical properties, and sensitivity of FOX-7 based PBXs via MD simulation

Author(s): Jianbo Fu, Baoguo Wang, Yafang Chen, Yunchuan Li, Xing Tan, Biyuan Wang, and Baoyun Ye,

This manuscript describes - Molecular dynamics (MD) simulations have been applied to investigate 1, 1-diamino-2, 2-dinitroethene (FOX-7) crystal and FOX-7(011)-based polymer bonded explosives (PBXs) with four typical polymers, polyethylene glycol (PEG), fluorine-polymer(F2603), ethylene-vinyl acetate copolymer (EVA) and ester urethane (ESTANE5703) under COMPASS force field. Binding energy (E_{bind}), cohesive energy density (CED), initiation bond length distribution, and isotropic mechanical properties of FOX-7 and its PBXs at different temperatures were reported for the first time. And the relationship between them and sensitivity.

In summary, this is very interesting work, However, I can not recommend acceptance to Royal Society Open Science without major revision of the paper according to the following comments.

Comment 1. p. 3, left column, line 4 from bottom - p.3, right column, line 22; (Simulation details MD simulations)

The temperature setting in this MD calculation is 400 K, but in order to clarify the impact sensitivity of high energetic materials, ideally it should be calculated at a temperature close to the detonation reaction. Therefore, the authors should extend the computational domain to at least 1000-2000K.

It is considered that the temperature set by the authors only reveals the thermal decomposition reaction.

Comment 2. p. 3, left column, line 4 from bottom - p.3, right column, line 22; (Simulation details MD simulations), p. 3, right column, line 23 from bottom - p.3, right column, bottom; (Results and discussion Judgment of equilibration),

Related to comment 1 above, if the temperature setting in the MD calculation is set to 1000-2000 K, it is extremely incompatible with the current calculation conditions from our experience.

Although the authors use the Verlet method for integrators, the energy conservation law described in the Judgment of equilibration (Results and discussion) section may not be guaranteed at 1000-2000 K.

First of all, the authors need to verify that the law of conservation of energy has been established reliably by changing the time step with the NVE ensemble as a test calculation before executing molecular dynamics simulations (NPT). If the authors attempt to simulate the relationship between FOX-7 crystal and FOX-7(011)-based PBXs with four typical polymers and sensitivity, the authors need to use the sixth symplectic integrator with a time step of 0.1 fs. the symplectic integrator is the numerical integration scheme for Hamiltonian system, which conserve the symplectic two-form exactly, so that $(q(0), p(0)) \rightarrow (q(\tau), p(\tau))$ is canonical transformation. This algorithm is accurate and has no accumulation of numerical errors for total energy in contrast to the other common algorithm to solve the Hamiltonian equation of motion. As a minimum consideration, it is necessary to confirm that the total energy is conserved when the simulation is executed with the time step of 0.1 fs to 0.01. From the result, the validity of the calculation result can not be guaranteed unless all calculations are performed again at the time step where the total energy is not conserved. With regard to the law of conservation of energy, the validation of the size of the supercell must also be shown to the reader.

Comment 3. p. 4, right column, line 6 - p.4, right column, line 8; (Results and discussion Binding energy)

What is the "hot point" theory? I know only "Hot Spot" theory. I couldn't get reference 24 (J. Xiao, L. Zhao, W. Zhu, J. Chen, G. Ji, F. Zhao, Q. Wu and H. Xiao, *Science China Chemistry*, 2012, 55, 2587-2594.). So, I didn't know the details of the "hot point" theory. This confuses the reader, so authors should elaborate on it.

Comment 4. p. 5, right column, line 3 - p.5, right column, bottom; (Results and discussion Initiation bond length distribution)

The authors should further investigate the phenomenon that the bond length of C-NO₂ in FOX-7 is increased by the addition of four polymers from the viewpoint of intermolecular interaction. Specifically, the intermolecular distance between C-NO₂ in FOX-7 and the four polymers should be shown to the reader. In addition, the differences in intermolecular interactions between the four polymers and FOX-7 should be discussed in detail. Readers are very dissatisfied with this point.

Decision letter (RSOS-200345.R0)

Dear Dr Wang:

Title: Computational analysis of energy, mechanical properties, and sensitivity of FOX-7 based PBXs via MD simulation
Manuscript ID: RSOS-200345

The editor assigned to your manuscript has now received comments from reviewers. We would like you to revise your paper in accordance with the referee and Subject Editor suggestions which can be found below (not including confidential reports to the Editor). Please note this decision does not guarantee eventual acceptance.

Please submit your revised paper before 19-Jun-2020. Please note that the revision deadline will expire at 00.00am on this date. If we do not hear from you within this time then it will be assumed that the paper has been withdrawn. In exceptional circumstances, extensions may be possible if agreed with the Editorial Office in advance. We do not allow multiple rounds of revision so we urge you to make every effort to fully address all of the comments at this stage. If deemed necessary by the Editors, your manuscript will be sent back to one or more of the original reviewers for assessment. If the original reviewers are not available we may invite new reviewers.

When submitting your revised manuscript, you must respond to the comments made by the referees and upload a file "Response to Referees" in "Section 6 - File Upload". Please use this to document how you have responded to the comments, and the adjustments you have made. In

order to expedite the processing of the revised manuscript, please be as specific as possible in your response.

On behalf of the Subject Editor Professor Anthony Stace and the Associate Editor Professor Kim Jelfs.

RSC Associate Editor:
Comments to the Author:
(There are no comments.)

RSC Subject Editor:
Comments to the Author:
(There are no comments.)

Reviewers' Comments to Author:
Reviewer: 1

Comments to the Author(s)

“shock wave sensitivity” should “shock sensitivity”; CL-20, RDX, HMX should be explained with full chemical name first; Codes A,B,C,D are not mentioned in caption of Figure 1; Captions of various figures such as Fig.8 is not below the figure; “This is since” should be “this is because”; there are much more clarity and grammar problems in the text;

Abstract and conclusion sections should be largely briefed with only key information;

When citing the literature, mentioning the family name is sufficient, e.g. “Vijaya S. MISHRA” could be briefed as “Vijaya”;

In Table 1, the experimental lattice parameters should be provided for FOX-7;

There is no relevant results presented on sensitivity of FOX-7 based PBXs, so the title of this paper is misleading;

Over the temperature of 300°C, the FOX-7 should undergo decomposition, and is it reasonable to investigate the mechanical properties under such high temperature?

At least one group of mechanical parameters should be compared with experimental ones, so that the models build in this paper could be validated;

The systems of P, P1 (only roughly shown in Figure 3) to P4 are not clear in structures, without providing any information on the assembling of the molecules, the repeating units of the polymers, the density of the packing, the total number of atoms for simulations;

The theoretical background of mechanical properties are well-known in the literature (equation 2 to 7 with relevant text), and should be moved to supporting information; there are some less important figures and tables could also be included in supporting information for overall simplicity and clarity; Similar results as Figure 4 for the other mixture systems should be provided in supporting information as well;

Reviewer: 2

Comments to the Author(s)

Journal: Royal Society Open Science

Manuscript ID RSOS-200345

Title: Computational analysis of energy, mechanical properties, and sensitivity of FOX-7 based PBXs via MD simulation

Author(s): Jianbo Fu, Baoguo Wang, Yafang Chen, Yunchuan Li, Xing Tan, Biyuan Wang, and Baoyun Ye,

This manuscript describes - Molecular dynamics (MD) simulations have been applied to investigate 1, 1-diamino-2, 2-dinitroethene (FOX-7) crystal and FOX-7(011)-based polymer bonded explosives (PBXs) with four typical polymers, polyethylene glycol (PEG), fluorine-polymer(F2603), ethylene-vinyl acetate copolymer (EVA) and ester urethane (ESTANE5703) under COMPASS force field. Binding energy (E_{bind}), cohesive energy density (CED), initiation bond length distribution, and isotropic mechanical properties of FOX-7 and its PBXs at different temperatures were reported for the first time. And the relationship between them and sensitivity.

In summary, this is very interesting work, However, I can not recommend acceptance to Royal Society Open Science without major revision of the paper according to the following comments.

Comment 1. p. 3, left column, line 4 from bottom - p.3, right column, line 22; (Simulation details MD simulations)

The temperature setting in this MD calculation is 400 K, but in order to clarify the impact sensitivity of high energetic materials, ideally it should be calculated at a temperature close to the detonation reaction. Therefore, the authors should extend the computational domain to at least 1000-2000K.

It is considered that the temperature set by the authors only reveals the thermal decomposition reaction.

Comment 2. p. 3, left column, line 4 from bottom - p.3, right column, line 22; (Simulation details MD simulations), p. 3, right column, line 23 from bottom - p.3, right column, bottom; (Results and discussion Judgment of equilibration),

Related to comment 1 above, if the temperature setting in the MD calculation is set to 1000-2000 K, it is extremely incompatible with the current calculation conditions from our experience. Although the authors use the Verlet method for integrators, the energy conservation law described in the Judgment of equilibration (Results and discussion) section may not be guaranteed at 1000-2000 K.

First of all, the authors need to verify that the law of conservation of energy has been established reliably by changing the time step with the NVE ensemble as a test calculation before executing molecular dynamics simulations (NPT). If the authors attempt to simulate the relationship between FOX-7 crystal and FOX-7(011)-based PBXs with four typical polymers and sensitivity, the authors need to use the sixth symplectic integrator with a time step of 0.1 fs. the symplectic integrator is the numerical integration scheme for Hamiltonian system, which conserve the symplectic two-form exactly, so that $(q(0), p(0)) \rightarrow (q(\tau), p(\tau))$ is canonical transformation. This algorithm is accurate and has no accumulation of numerical errors for total energy in contrast to the other common algorithm to solve the Hamiltonian equation of motion. As a minimum

consideration, it is necessary to confirm that the total energy is conserved when the simulation is executed with the time step of 0.1 fs to 0.01. From the result, the validity of the calculation result can not be guaranteed unless all calculations are performed again at the time step where the total energy is not conserved. With regard to the law of conservation of energy, the validation of the size of the supercell must also be shown to the reader.

Comment 3. p. 4, right column, line 6 - p.4, right column, line 8; (Results and discussion Binding energy)

What is the "hot point" theory? I know only "Hot Spot" theory.

I couldn't get reference 24 (J. Xiao, L. Zhao, W. Zhu, J. Chen, G. Ji, F. Zhao, Q. Wu and H. Xiao, Science China Chemistry, 2012, 55, 2587-2594.). So, I didn't know the details of the "hot point" theory. This confuses the reader, so authors should elaborate on it.

Comment 4. p. 5, right column, line 3 - p.5, right column, bottom; (Results and discussion Initiation bond length distribution)

The authors should further investigate the phenomenon that the bond length of C-NO₂ in FOX-7 is increased by the addition of four polymers from the viewpoint of intermolecular interaction. Specifically, the intermolecular distance between C-NO₂ in FOX-7 and the four polymers should be shown to the reader. In addition, the differences in intermolecular interactions between the four polymers and FOX-7 should be discussed in detail. Readers are very dissatisfied with this point.

Author's Response to Decision Letter for (RSOS-200345.R0)

See Appendix A.

RSOS-200345.R1 (Revision)

Review form: Reviewer 2

Is the manuscript scientifically sound in its present form?

Yes

Are the interpretations and conclusions justified by the results?

Yes

Is the language acceptable?

Yes

Do you have any ethical concerns with this paper?

No

Have you any concerns about statistical analyses in this paper?

No

Recommendation?

Major revision is needed (please make suggestions in comments)

Comments to the Author(s)

Journal: Royal Society Open Science

Manuscript ID RSOS-200345.R1

Title: Computational analysis the relationships of energy and mechanical properties with sensitivity for FOX-7 based PBXs via MD simulation

Author(s): Jianbo Fu, Baoguo Wang, Yafang Chen, Yunchuan Li, Xing Tan, Biyuan Wang, and Baoyun Ye,

This manuscript describes – Molecular dynamics (MD) simulations have been applied to investigate 1, 1-diamino-2, 2-dinitroethene (FOX-7) crystal and FOX-7(011)-based polymer bonded explosives (PBXs) with four typical polymers, polyethylene glycol (PEG), fluorine-polymer(F2603), ethylene-vinyl acetate copolymer (EVA) and ester urethane(ESTANE5703) under COMPASS force field. Binding energy (E_{bind}), cohesive energy density (CED), initiation bond length distribution, radial distribution function, and isotropic mechanical properties of FOX-7 and its PBXs at different temperatures were reported for the first time and the relationship between them and sensitivity.

The reviewers found that the author made efforts to improve the manuscript significantly. However, I can not recommend acceptance to Royal Society Open Science without major revision of the paper according to the following Re-Comment 4.

Comment 4. p. 5, right column, line 3 - p.5, right column, bottom; (Results and discussion Initiation bond length distribution)

The authors should further investigate the phenomenon that the bond length of C-NO₂ in FOX-7 is increased by the addition of four polymers from the viewpoint of intermolecular interaction. Specifically, the intermolecular distance between C-NO₂ in FOX-7 and the four polymers should be shown to the reader. In addition, the differences in intermolecular interactions between the four polymers and FOX-7 should be discussed in detail. Readers are very dissatisfied with this point.

Author's Response; Your suggestions are very useful. Regarding the interaction between FOX-7 and the four polymers, I added a new chapter "Radial Distribution Function" to the new manuscript. This chapter analyzes in detail the interaction between FOX-7 and four polymers. You can view it on the ninth page of the article. Thank you again for your suggestions and hope to learn more from you.

Re-Comment 4.p. 6, left column, line 9 from bottom - p.6, right column, line 2; (Results and discussion Initiation bond length distribution), p. 8, right column, top - p.9, right column, line 4 from bottom; (Results and discussion Radial distribution function)

First of all, what does "The radial distribution function ginter (r)" mean? I only know "The radial distribution function (r)". Please elaborate on the meaning of "ginter" to the reader.

After molecular dynamics calculations, radial distribution functions are usually used to discuss the details of intermolecular structure. Acknowledging that, there are some questions.

1. What do the symbols a, b, c in Figure 16 mean? Corresponding to the text of the authors, it is thought that a represents hydrogen bond and b represents van der Waals force. Then, what does c correspond to in Figure S10, 11, and 12? · · Authors should kindly explain to the reader.

2. In the description in the text of Fig. 16, "at low temperatures(198-298K), hydrogen bonds exist between H1-O2 molecules, but as the temperature increases, the hydrogen bonds gradually disappear." However, can Figure 16 really reasonably lead to such a conclusion? ·· I'm curious about the new peaks that are appearing at close range. Similarly, there are explanations in various places that I cannot understand. The text of this session will only confuse the reader. The authors should explain in an easy-to-read manner to the reader.

3. In comment 4, I commented that it is necessary to investigate the phenomenon that the bond length of C-NO₂ in FOX-7 is increased by the addition of four polymers from the viewpoint of intermolecular interaction. Did you know anything about this with the radial distribution function? Authors should explain to readers in detail.

In addition, the differences in intermolecular interactions between the four polymers and FOX-7 need to be elaborated.

Decision letter (RSOS-200345.R1)

Dear Dr Wang:

Title: Computational analysis the relationships of energy and mechanical properties with sensitivity for FOX-7 based PBXs via MD simulation
Manuscript ID: RSOS-200345.R1

The editor assigned to your paper has now received comments from reviewers. We would like you to revise your paper in accordance with the referee and Subject Editor suggestions which can be found below (not including confidential reports to the Editor). Please note this decision does not guarantee eventual acceptance.

Please submit a copy of your revised paper before 06-Aug-2020. Please note that the revision deadline will expire at 00.00am on this date. If we do not hear from you within this time then it will be assumed that the paper has been withdrawn. In exceptional circumstances, extensions may be possible if agreed with the Editorial Office in advance. We do not allow multiple rounds of revision so we urge you to make every effort to fully address all of the comments at this stage. If deemed necessary by the Editors, your manuscript will be sent back to one or more of the original reviewers for assessment. If the original reviewers are not available we may invite new reviewers.

On behalf of the Subject Editor Professor Anthony Stace and the Associate Editor Professor Kim Jelfs.

RSC Associate Editor:
Comments to the Author:
(There are no comments.)

RSC Subject Editor:
Comments to the Author:
(There are no comments.)

Reviewers' Comments to Author:
Reviewer: 2

Comments to the Author(s)
Journal: Royal Society Open Science
Manuscript ID RSOS-200345.R1
Title: Computational analysis the relationships of energy and mechanical properties with sensitivity for FOX-7 based PBXs via MD simulation

Author(s): Jianbo Fu, Baoguo Wang, Yafang Chen, Yunchuan Li, Xing Tan, Biyuan Wang, and Baoyun Ye,

This manuscript describes – Molecular dynamics (MD) simulations have been applied to investigate 1, 1-diamino-2, 2-dinitroethene (FOX-7) crystal and FOX-7(011)-based polymer bonded explosives (PBXs) with four typical polymers, polyethylene glycol (PEG), fluorine-polymer(F2603), ethylene-vinyl acetate copolymer (EVA) and ester urethane(ESTANE5703) under COMPASS force field. Binding energy (Ebind), cohesive energy density (CED), initiation bond length distribution, radial distribution function, and isotropic mechanical properties of FOX-7

and its PBXs at different temperatures were reported for the first time and the relationship between them and sensitivity.

The reviewers found that the author made efforts to improve the manuscript significantly. However, I can not recommend acceptance to Royal Society Open Science without major revision of the paper according to the following Re-Comment 4.

Comment 4. p. 5, right column, line 3 - p.5, right column, bottom; (Results and discussion Initiation bond length distribution)

The authors should further investigate the phenomenon that the bond length of C-NO₂ in FOX-7 is increased by the addition of four polymers from the viewpoint of intermolecular interaction. Specifically, the intermolecular distance between C-NO₂ in FOX-7 and the four polymers should be shown to the reader. In addition, the differences in intermolecular interactions between the four polymers and FOX-7 should be discussed in detail. Readers are very dissatisfied with this point.

Author's Response; Your suggestions are very useful. Regarding the interaction between FOX-7 and the four polymers, I added a new chapter "Radial Distribution Function" to the new manuscript. This chapter analyzes in detail the interaction between FOX-7 and four polymers. You can view it on the ninth page of the article. Thank you again for your suggestions and hope to learn more from you.

Re-Comment 4.p. 6, left column, line 9 from bottom - p.6, right column, line 2; (Results and discussion Initiation bond length distribution), p. 8, right column, top - p.9, right column, line 4 from bottom; (Results and discussion Radial distribution function)

First of all, what does "The radial distribution function ginter (r)" mean? I only know "The radial distribution function (r)". Please elaborate on the meaning of "ginter" to the reader.

After molecular dynamics calculations, radial distribution functions are usually used to discuss the details of intermolecular structure. Acknowledging that, there are some questions.

1. What do the symbols a, b, c in Figure 16 mean? Corresponding to the text of the authors, it is thought that a represents hydrogen bond and b represents van der Waals force. Then, what does c correspond to in Figure S10, 11, and 12? · · Authors should kindly explain to the reader.

2. In the description in the text of Fig. 16, "at low temperatures(198-298K), hydrogen bonds exist between H1-O2 molecules, but as the temperature increases, the hydrogen bonds gradually disappear." However, can Figure 16 really reasonably lead to such a conclusion? · · I'm curious about the new peaks that are appearing at close range. Similarly, there are explanations in various places that I cannot understand. The text of this session will only confuse the reader. The authors should explain in an easy-to-read manner to the reader.

3. In comment 4, I commented that it is necessary to investigate the phenomenon that the bond length of C-NO₂ in FOX-7 is increased by the addition of four polymers from the viewpoint of intermolecular interaction. Did you know anything about this with the radial distribution function? Authors should explain to readers in detail.

In addition, the differences in intermolecular interactions between the four polymers and FOX-7 need to be elaborated.

Author's Response to Decision Letter for (RSOS-200345.R1)

See Appendix B.

RSOS-200345.R2 (Revision)

Review form: Reviewer 2

Is the manuscript scientifically sound in its present form?

Yes

Are the interpretations and conclusions justified by the results?

Yes

Is the language acceptable?

Yes

Do you have any ethical concerns with this paper?

No

Have you any concerns about statistical analyses in this paper?

No

Recommendation?

Major revision is needed (please make suggestions in comments)

Comments to the Author(s)

Journal: Royal Society Open Science

Manuscript ID RSOS-200345.R2

Title: Computational analysis the relationships of energy and mechanical properties with sensitivity for FOX-7 based PBXs via MD simulation

Author(s): Jianbo Fu, Baoguo Wang, Yafang Chen, Yunchuan Li, Xing Tan, Biyuan Wang, and Baoyun Ye,

This manuscript describes - Molecular dynamics (MD) simulations have been applied to investigate 1, 1-diamino-2, 2-dinitroethene (FOX-7) crystal and FOX-7(011)-based polymer bonded explosives (PBXs) with four typical polymers, polyethylene glycol (PEG), fluorine-polymer(F2603), ethylene-vinyl acetate copolymer (EVA) and ester urethane(ESTANE5703) under COMPASS force field. Binding energy (Ebind), cohesive energy density (CED), initiation bond length distribution, radial distribution function, and isotropic mechanical properties of FOX-7 and its PBXs at different temperatures were reported for the first time and the relationship between them and sensitivity.

In summary, this is very interesting work. However, I can not recommend acceptance to Royal Society Open Science without major revision of the paper according to the following comments.

Comment 5. p. 7, right column, line 17 from bottom - p.8, left column, line 9 (It is shown in red letters.), Fig. 15, Table 5; (Results and discussion Initiation bond length distribution)

The authors described that the hydrogen bonding and van der Waals forces of P1 are the highest compared to other systems. However, objectively, P1 and P2 are almost the same. The Lave value for P1 is minimal, but the "significant" Lmax is the second height. The authors described that Lmax is a more critical parameter than Lave.

In general, even in advanced quantum chemical calculations, discussion of the bond distance with a difference of 3 ~ 4 digits or less after the decimal point causes a problem of calculation accuracy.

Therefore, it is not convincing to the reader that molecular dynamics calculation, which is not quantum chemical calculation, advances the discussion with a difference of 3 ~ 4 decimal places or less in the bond distance.

In addition, the authors presume that the large interaction between ESTANE5703 and EVA is due to the high number of 140-180° hydrogen bonds. Generally, the strength of hydrogen bonds depends on the distance between the bonding atoms and the distribution of charges. The bond angle is only a secondary effect. The authors should show and explain the data to the reader.

Comment 6. p. 6, left column, line 9 from bottom - p.7, right column, line 2, (Results and discussion Initiation bond length distribution)

As already mentioned in Comment 5, it is not convincing to the reader that molecular dynamics calculation, which is not quantum chemical calculation, advances the discussion due to the difference of the bonding distance to 3 ~ 4 decimal places or less. Therefore, this session will need to be significantly rewritten for the reader to understand.

Comment 7. p. 9, right column, line 10 from bottom - p.10, left column, bottom (It is shown in red letters.), Fig. 17; (Results and discussion Radial distribution function)

First of all, Fig. 17 seems to be P1, but I don't know which P2 to P4 corresponds to Fig. S10-12. This point is very unfriendly to the reader. Please explain the figure so that the reader can understand it. The sentences in the red part are very difficult for the reader to understand. The authors describe - "Hydrogen bonding mainly exists in O-H atom pairs and F-H atom pairs, and there is almost no hydrogen bonding in N-H atom pairs." However, in Figure 17(B), hydrogen bonds are present in N1-H2. Further, only H1-O2 (A), N1-H2 (B) and O1-H2 (C) are shown in Fig. 17 and Fig. S10-12. The reader cannot judge the hydrogen bond of the F-H atom pairs. The authors describe - "Through comparative analysis, it is found that the interaction between FOX-7 and PEG is stronger than other systems, which is reflected in that O1-H2 and N1-H2 have the highest peak values of hydrogen bonds and strong van der Waals forces in the four systems at different temperatures." However, "is there almost no hydrogen bonding in N-H atom pairs?" From Fig. 17 and Fig. S10-12, I can not understand that the interaction between FOX-7 and PEG is clearly stronger than in other systems. The authors should improve the presentation of the figure to show the reader that the interaction between FOX-7 and PEG is stronger than other systems. The authors need to rewrite the radial distribution function diagram significantly to make it easier for the reader.

Review form: Reviewer 3

Is the manuscript scientifically sound in its present form?

No

Are the interpretations and conclusions justified by the results?

No

Is the language acceptable?

Yes

Do you have any ethical concerns with this paper?

No

Have you any concerns about statistical analyses in this paper?

No

Recommendation?

Major revision is needed (please make suggestions in comments)

Comments to the Author(s)

This is the review for Manuscript RSOS-200345.R2 by Fu, Jianbo et. al. titled "Computational analysis the relationships of energy and mechanical properties with sensitivity for FOX-7 based PBXs via MD simulation"

The authors report results of several MD calculations of 1,1-diamino-2,2-dinitroethane - based polymer blends, to investigate how different materials can decrease the shock sensitivity of the resulting explosive blends. The authors made several improvements on the previous versions of the manuscript, and the work is of practical value. However, I recommend the following major changes before this work can be published.

- a) There should be no acronyms in the title. MD, FOX-7 and PBXs should be spelled out. The title should be shorter. I suggest something like "Exploring shock sensitivity of high energy materials with molecular dynamics"
- b) In the construction of polymer models, please explain the "Smart" method the "Anderson" thermostat and the "Fine" quality for those readers unfamiliar with COMPASS and whatever other software the authors are using.
- c) A mention of the mechanism by which FOX-7 explodes would be very valuable. I am confused, in particular, by the very elevated high-temperature simulations. I though polynitrated compounds explode via radical branching not via thermochemical acceleration. In fact it may be the case that at different Ts one mechanism prevails over the other.
- d) In the initiation bond length distribution: Figure 11 is too busy. One bond length distribution is probably sufficient to make the point.
- e) In the same section, how is L_{max} calculated?
- f) In the conclusion section the authors talk about the binding energy. This needs to be clarified. The binding energies of what? Relative to what?
- g) Finally, and most importantly, the authors need to do a better job at convincing the reader that the mechanical and structural properties they compute do in fact explain how certain polymer blends do a better job at decreasing the sensitivity of high energy materials. Is there evidence in the literature that the authors approach actually works?
How do their results compare with experiment? To me, without a thorough understanding of the mechanism and a better characterization of the force fields with some barriers (transition states) is

not sufficient evidence that the various materials considered actually do decrease the sensitivity of FOX-7.

Decision letter (RSOS-200345.R2)

Dear Dr Wang:

Title: Computational analysis the relationships of energy and mechanical properties with sensitivity for FOX-7 based PBXs via MD simulation
Manuscript ID: RSOS-200345.R2

The editor assigned to your paper has now received comments from reviewers. We would like you to revise your paper in accordance with the referee and Subject Editor suggestions which can be found below (not including confidential reports to the Editor). Please note this decision does not guarantee eventual acceptance.

Please submit a copy of your revised paper before 25-Nov-2020. Please note that the revision deadline will expire at 00.00am on this date. If we do not hear from you within this time then it will be assumed that the paper has been withdrawn. In exceptional circumstances, extensions may be possible if agreed with the Editorial Office in advance. We do not allow multiple rounds of revision so we urge you to make every effort to fully address all of the comments at this stage. If deemed necessary by the Editors, your manuscript will be sent back to one or more of the original reviewers for assessment. If the original reviewers are not available we may invite new reviewers.

On behalf of the Subject Editor Professor Anthony Stace and the Associate Editor Professor Kim Jelfs.

RSC Associate Editor:
 Comments to the Author:
 (There are no comments.)

RSC Subject Editor:
 Comments to the Author:
 (There are no comments.)

Reviewers' Comments to Author:
 Reviewer: 2

Comments to the Author(s)
 Journal: Royal Society Open Science
 Manuscript ID RSOS-200345.R2
 Title: Computational analysis the relationships of energy and mechanical properties with sensitivity for FOX-7 based PBXs via MD simulation

Author(s): Jianbo Fu, Baoguo Wang, Yafang Chen, Yunchuan Li, Xing Tan, Biyuan Wang, and Baoyun Ye,

This manuscript describes - Molecular dynamics (MD) simulations have been applied to investigate 1, 1-diamino-2, 2-dinitroethene (FOX-7) crystal and FOX-7(011)-based polymer bonded explosives (PBXs) with four typical polymers, polyethylene glycol (PEG), fluorine-polymer(F2603), ethylene-vinyl acetate copolymer (EVA) and ester urethane(ESTANE5703) under COMPASS force field. Binding energy (Ebind), cohesive energy density (CED), initiation bond length distribution, radial distribution function, and isotropic mechanical properties of FOX-7 and its PBXs at different temperatures were reported for the first time and the relationship between them and sensitivity.

In summary, this is very interesting work. However, I can not recommend acceptance to Royal Society Open Science without major revision of the paper according to the following comments.

Comment 5. p. 7, right column, line 17 from bottom - p.8, left column, line 9 (It is shown in red letters.), Fig. 15, Table 5; (Results and discussion Initiation bond length distribution)

The authors described that the hydrogen bonding and van der Waals forces of P1 are the highest compared to other systems. However, objectively, P1 and P2 are almost the same. The Lave value for P1 is minimal, but the "significant" Lmax is the second height. The authors described that Lmax is a more critical parameter than Lave.

In general, even in advanced quantum chemical calculations, discussion of the bond distance with a difference of 3 ~ 4 digits or less after the decimal point causes a problem of calculation accuracy.

Therefore, it is not convincing to the reader that molecular dynamics calculation, which is not quantum chemical calculation, advances the discussion with a difference of 3 ~ 4 decimal places or less in the bond distance.

In addition, the authors presume that the large interaction between ESTANE5703 and EVA is due to the high number of 140-180° hydrogen bonds. Generally, the strength of hydrogen bonds depends on the distance between the bonding atoms and the distribution of charges. The bond angle is only a secondary effect. The authors should show and explain the data to the reader.

Comment 6. p. 6, left column, line 9 from bottom - p.7, right column, line 2, (Results and discussion Initiation bond length distribution)

As already mentioned in Comment 5, it is not convincing to the reader that molecular dynamics calculation, which is not quantum chemical calculation, advances the discussion due to the difference of the bonding distance to 3 ~ 4 decimal places or less. Therefore, this session will need to be significantly rewritten for the reader to understand.

Comment 7. p. 9, right column, line 10 from bottom - p.10, left column, bottom (It is shown in red letters.), Fig. 17; (Results and discussion Radial distribution function)

First of all, Fig. 17 seems to be P1, but I don't know which P2 to P4 corresponds to Fig. S10-12. This point is very unfriendly to the reader. Please explain the figure so that the reader can understand it. The sentences in the red part are very difficult for the reader to understand. The authors describe - "Hydrogen bonding mainly exists in O-H atom pairs and F-H atom pairs, and there is almost no hydrogen bonding in N-H atom pairs." However, in Figure 17(B), hydrogen bonds are present in N1-H2. Further, only H1-O2 (A), N1-H2 (B) and O1-H2 (C) are shown in Fig. 17 and Fig. S10-12. The reader cannot judge the hydrogen bond of the F-H atom pairs. The authors describe - "Through comparative analysis, it is found that the interaction between FOX-7 and PEG is stronger than other systems, which is reflected in that O1-H2 and N1-H2 have the highest peak values of hydrogen bonds and strong van der Waals forces in the four systems at different temperatures." However, "is there almost no hydrogen bonding in N-H atom pairs?" From Fig. 17 and Fig. S10-12, I can not understand that the interaction between FOX-7 and PEG is clearly stronger than in other systems. The authors should improve the presentation of the figure to show the reader that the interaction between FOX-7 and PEG is stronger than other systems. The authors need to rewrite the radial distribution function diagram significantly to make it easier for the reader.

Reviewer: 3

Comments to the Author(s)

This is the review for Manuscript RSOS-200345.R2 by Fu, Jianbo et. al. titled "Computational analysis the relationships of energy and mechanical properties with sensitivity for FOX-7 based PBXs via MD simulation"

The authors report results of several MD calculations of 1,1-diamino-2,2-dinitroethane - based polymer blends, to investigate how different materials can decrease the shock sensitivity of the resulting explosive blends. The authors made several improvements on the previous versions of the manuscript, and the work is of practical value. However, I recommend the following major changes before this work can be published.

a) There should be no acronyms in the title. MD, FOX-7 and PBXs should be spelled out. The title should be shorter. I suggest something like "Exploring shock sensitivity of high energy materials with molecular dynamics"

b) In the construction of polymer models, please explain the "Smart" method the "Anderson" thermostat and the "Fine" quality for those readers unfamiliar with COMPASS and whatever other software the authors are using.

c) A mention of the mechanism by which FOX-7 explodes would be very valuable. I am confused, in particular, by the very elevated high-temperature simulations. I thought polynitrated compounds explode via radical branching not via thermochemical acceleration. In fact it may be the case that at different Ts one mechanism prevails over the other.

d) In the initiation bond length distribution: Figure 11 is too busy. One bond length distribution is probably sufficient to make the point.

e) In the same section, how is L_{max} calculated?

f) In the conclusion section the authors talk about the binding energy. This needs to be clarified. The binding energies of what? Relative to what?

g) Finally, and most importantly, the authors need to do a better job at convincing the reader that the mechanical and structural properties they compute do in fact explain how certain polymer blends do a better job at decreasing the sensitivity of high energy materials. Is there evidence in the literature that the authors approach actually works? How do their results compare with experiment? To me, without a thorough understanding of the mechanism and a better characterization of the force fields with some barriers (transition states) is not sufficient evidence that the various materials considered actually do decrease the sensitivity of FOX-7.

Author's Response to Decision Letter for (RSOS-200345.R2)

See Appendix C.

RSOS-200345.R3 (Revision)

Review form: Reviewer 2

Is the manuscript scientifically sound in its present form?

Yes

Are the interpretations and conclusions justified by the results?

Yes

Is the language acceptable?

Yes

Do you have any ethical concerns with this paper?

Yes

Have you any concerns about statistical analyses in this paper?

Yes

Recommendation?

Accept as is

Comments to the Author(s)

Journal: Royal Society Open Science

Manuscript ID RSOS-200345.R3

Title: Computational analysis the relationships of energy and mechanical properties with sensitivity for FOX-7 based PBXs via MD simulation

Author(s): Jianbo Fu, Baoguo Wang, Yafang Chen, Yunchuan Li, Xing Tan, Biyuan Wang, and Baoyun Ye,

This manuscript describes - Molecular dynamics (MD) simulations have been applied to investigate 1, 1-diamino-2, 2-dinitroethene (FOX-7) crystal and FOX-7(011)-based polymer bonded explosives (PBXs) with four typical polymers, polyethylene glycol (PEG), fluorine-polymer(F2603), ethylene-vinyl acetate copolymer (EVA) and ester urethane (ESTANE5703) under COMPASS force field. Binding energy (E_{bind}), cohesive energy density (CED), initiation bond length distribution, radial distribution function, and isotropic mechanical properties of FOX-7 and its PBXs at different temperatures were reported for the first time and the relationship between them and sensitivity.

Recommendation: The reviewer found that the authors sincerely answered the questions of the comments. Several points were still subjected to further revisions, but they were rather minor. In conclusion, the revised manuscript might be publishable.

Decision letter (RSOS-200345.R3)

This year has been very difficult for everyone, and we want to take the opportunity to thank you for your continued support in 2020.

The Royal Society Open Science editorial office will be closed from the evening of Friday 18 December 2020 until Monday 4 January 2021. We will not be responding during this time. If you have received a deadline within this time period, please contact us as soon as possible to allow us to extend the deadline. If you receive any automated messages during this time asking you to meet a deadline, we offer apologies and invite you to respond after the festive period or during normal working hours.

With our best for a peaceful festive period and New Year, and we look forward to working with you in 2021.

Dear Dr Wang:

Title: Computational analysis the relationships of energy and mechanical properties with sensitivity for FOX-7 based PBXs via MD simulation

Manuscript ID: RSOS-200345.R3

It is a pleasure to accept your manuscript in its current form for publication in Royal Society Open Science. The chemistry content of Royal Society Open Science is published in collaboration with the Royal Society of Chemistry.

On behalf of the Subject Editor Professor Anthony Stace and the Associate Editor Professor Kim Jelfs.

RSC Associate Editor:
Comments to the Author:
(There are no comments.)

RSC Subject Editor:
Comments to the Author:
(There are no comments.)

Reviewer(s)' Comments to Author:
Reviewer: 2

Comments to the Author(s)
Journal: Royal Society Open Science
Manuscript ID RSOS-200345.R3
Title: Computational analysis of the relationships of energy and mechanical properties with sensitivity for FOX-7 based PBXs via MD simulation

Author(s): Jianbo Fu, Baoguo Wang, Yafang Chen, Yunchuan Li, Xing Tan, Biyuan Wang, and Baoyun Ye,

This manuscript describes - Molecular dynamics (MD) simulations have been applied to investigate 1, 1-diamino-2, 2-dinitroethene (FOX-7) crystal and FOX-7(011)-based polymer bonded explosives (PBXs) with four typical polymers, polyethylene glycol (PEG), fluorine-polymer(F2603), ethylene-vinyl acetate copolymer (EVA) and ester urethane (ESTANE5703)

under COMPASS force field. Binding energy (E_{bind}), cohesive energy density (CED), initiation bond length distribution, radial distribution function, and isotropic mechanical properties of FOX-7 and its PBXs at different temperatures were reported for the first time and the relationship between them and sensitivity.

Recommendation: The reviewer found that the authors sincerely answered the questions of the comments. Several points were still subjected to further revisions, but they were rather minor. In conclusion, the revised manuscript might be publishable.

Appendix A

Dear Editors and Reviewers:

We thank the two editors and the two reviewers for giving this manuscript the opportunity to be revised and for suggesting revisions. All of your suggestions are very important and they will guide me in my future research work. Below I will list all the manuscript revisions, as well as the responses to each reviewer's suggestions and questions.

List of Action:

LOA1: We extended the simulated temperature range to 1000-1400K. An energy conservation comparison was performed between the Verlet velocity integrator and the sixth order symplectic integrator (NVE ensemble, 0.1 steps). The sixth order symplectic integrator was used to verify the conservation of energy under the NVE ensemble, and the NPT-MD was performed again. It can be viewed in the last paragraph of the "MD simulations" section, in the first paragraph of the "Judgment of equilibration" section of the new manuscript and in the supplement information S.1.

LOA2: Added a detailed list of polymer model construction. It mainly provides the repeating unit, the number of segments, the packing density and the total number of atoms of the four polymer models. It can be viewed in Table1 in the new manuscript.

LOA3: The change rule of CED is analyzed at high temperatures (1000-1400K). The change law is consistent with the low temperatures (198-398K). It can be viewed in the last paragraph of the "Cohesive Energy Density" section in the new manuscript.

LOA4: The change rule of bond length distribution is analyzed at high temperatures (1000-1400K). The impact sensitivity of four FOX-7 based PBXs was compared. It can be viewed in the last paragraph of the "Initiation bond length distribution" section in the new manuscript.

LOA5: Added a set of tensile strength experiments. By comparing the experimental results with the simulation results, the effectiveness of the simulation experiments is verified. The experimental results can be viewed in the second paragraph of the "Mechanical properties" section of the new manuscript, and the experimental process can be viewed in S.3.2 of the supplement information.

LOA6: The basic theory of mechanical properties is transferred from the main article to supplement information. It can be viewed in S.3.1 of the supplement information.

LOA7: To reveal the interaction between FOX-7 and the four polymers, the analysis of radial function distribution was supplemented. The effects of hydrogen bonding and van der Waals forces at different temperatures are analyzed. It can be viewed in the "Radial distribution function" section of the new manuscript.

LOA8: The abstract and conclusion have been simplified. It can be viewed in the "Abstract" and "Conclusion" of the new manuscript.

LOA9: The detailed data of the binding energy is transferred from the main article to the supplement information. It can be viewed in Table S1 in the supplement information.

LOA10: The article title was changed from "Computational analysis of energy, mechanical properties, and sensitivity of FOX-7 based PBXs via MD simulation" to "Computational analysis the relationships of energy and mechanical properties with sensitivity for FOX-7 based PBXs via MD simulation".

Responses to Reviewers

To Reviewer 1:

Comment 1. " shock wave sensitivity" should " shock sensitivity" ; CL-20, RDX, HMX should be explained with full chemical name first; Codes A,B,C,D are not mentioned in caption of Figure 1; Captions of various figures such as Fig.8 is not below the figure; " This is since" should be " this is because" ; There are much more clarity and grammar problems in the text;

Thank you for your comments regarding grammatical and writing errors. These issues we have completed fixing in the new manuscript. (1) The amendment was completed at the end of the second paragraph in the "Introduction". (2) An amendment was completed in the middle of the third paragraph in the "Introduction". (3) Already modified in the note under Figure 1. (4) The problem of misalignment of illustrations occurred after the manuscript was uploaded, and this problem did not appear in the manuscript that was not uploaded. (5) Complete the amendment

in the middle of the first paragraph in the "Choice of force field". (6) The full text has been re-written checked and revised.

Comment 2. Abstract and conclusion sections should be largely briefed with only key information;

The abstract and conclusion have been simplified in the new manuscript.

Comment 3. When citing the literature, mentioning the family name is sufficient, e.g. "Vijaya S. MISHRA" could be briefed as "Vijaya" ;

It has been modified according to your suggestion. It can be viewed in the second paragraph of the "Introduction" section in the new manuscript.

Comment 4. In Table 1, the experimental lattice parameters should be provided for FOX-7;

The experimental lattice parameters have been provided in Table 2 in the new manuscript.

Comment 5. There is no relevant results presented on sensitivity of FOX-7 based PBXs, so the title of this paper is misleading;

The title of the article is indeed misleading, we have revised the title, see **LOA10**.

Comment 6. Over the temperature of 300 °C, the FOX-7 should undergo decomposition, and is it reasonable to investigate the mechanical properties under such high temperature?

The discussion of the mechanical properties of FOX-7 based PBX in this article did not exceed 300 °C, but was conducted under 198-398K (-75 -125 °C). There are two reasons for choosing this temperature range:

1. Due to the increasing complexity of modern battlefield environments, discussing the mechanical properties of PBX at different temperatures (198-398K) can reflect its ability to adapt to diverse battlefield environments. It plays a positive role in the screening and safe use of PBX formula.

2. As the temperature increases, PBX becomes more sensitive, and its mechanical properties also change. The relationship between mechanical properties and sensitivity is discussed by observing the law of mechanical properties changing with temperature. You can refer to this document: Molecular dynamics simulations on 3-CL-20-based PBXs with added GAP and its derivative polymers. RSC Advances (DOI: 10.1039/c7ra13517c).

Comment 7. At least one group of mechanical parameters should be compared with experimental ones, so that the models build in this paper could be validated;

We have taken your suggestions and added a set of tensile strength experiments. The experimental results can be viewed in the second paragraph of the "Mechanical properties" section of the new manuscript, and the experimental process can be viewed in S.3.2 of the supplement information.

Comment 8. The systems of P, P1 (only roughly shown in Figure 3) to P4 are not clear in structures, without providing any information on the assembling of the molecules, the repeating units of the polymers, the density of the packing, the total number of atoms for simulations;

It is our oversight to fail to provide relevant details. Table 1 in the new manuscript supplemented the details of the polymer model construction.

Comment 9. The theoretical background of mechanical properties are well-known in the literature (equation 2

to 7 with relevant text), and should be moved to supporting information; there are some less important figures and tables could also be included in supporting information for overall simplicity and clarity; Similar results as Figure 4 for the other mixture systems should be provided in supporting information as well;

Based on your suggestions, we have transferred the theoretical background of mechanical properties and the detailed data sheet of binding energy to the supplementary information. It can be viewed in S.2 and S.3.1 of the supplementary information. In the new manuscript, Fig. 4 is changed to Fig. 5, and we have shown similar results for other mixture systems in S.1 with supplement information.

Thank you again for your suggestions and hope to learn more from you.

To Reviewer 2:

Comment 1. p. 3, left column, line 4 from bottom - p.3, right column, line 22; (Simulation details MD simulations)
The temperature setting in this MD calculation is 400 K, but in order to clarify the impact sensitivity of high energetic materials, ideally it should be calculated at a temperature close to the detonation reaction. Therefore, the authors should extend the computational domain to at least 1000-2000K.

It is considered that the temperature set by the authors only reveals the thermal decomposition reaction.

Comment 2. p. 3, left column, line 4 from bottom - p.3, right column, line 22; (Simulation details MD simulations), p. 3, right column, line 23 from bottom - p.3, right column, bottom; (Results and discussion Judgment of equilibration),

Related to comment 1 above, if the temperature setting in the MD calculation is set to 1000-2000 K, it is extremely incompatible with the current calculation conditions from our experience. Although the authors use the Verlet method for integrators, the energy conservation law described in the Judgment of equilibration (Results and discussion) section may not be guaranteed at 1000-2000 K.

First of all, the authors need to verify that the law of conservation of energy has been established reliably by changing the time step with the NVE ensemble as a test calculation before executing molecular dynamics simulations (NPT). If the authors attempt to simulate the relationship between FOX-7 crystal and FOX-7(O11)-based PBXs with four typical polymers and sensitivity, the authors need to use the sixth symplectic integrator with a time step of 0.1 fs. the symplectic integrator is the numerical integration scheme for Hamiltonian system, which conserve the symplectic two-form exactly, so that $(q(0), p(0)) \rightarrow (q(\tau), p(\tau))$ is canonical transformation. This algorithm is accurate and has no accumulation of numerical errors for total energy in contrast to the other common algorithm to solve the Hamiltonian equation of motion. As a minimum consideration, it is necessary to confirm that the total energy is conserved when the simulation is executed with the time step of 0.1 fs to 0.01. From the result, the validity of the calculation result can not be guaranteed unless all calculations are performed again at the time step where the total energy is conserved. With regard to the law of conservation of energy, the validation of the size of the supercell must also be shown to the reader.

Thank you for your valuable suggestions. Since comment 1 and comment 2 are related, reply here together. We have selected five temperatures of 1000K, 1100K, 1200K, 1300K, and 1400K from the 1000-2000K interval you suggested to conduct molecular dynamics simulation again. The energy conservation of the Verlet velocity integrator and the sixth order symplectic integrator were compared at high temperatures. The result of the comparison experiment is consistent with your suggestion. In the new manuscript we have added MD simulation at high temperatures. The simulation process, verification of supercell size and energy balance can be viewed in the last paragraph of the "MD simulations" section of the new manuscript, the first paragraph of the "Judgment of equilibration" section, and supplement information S.1. The change rule of CED at high temperatures (1000-1400K) can be viewed in the last paragraph of the "Cohesive energy density" section in the new manuscript. The change of bond length distribution induced by high temperature (1000-1400K) and comparison of the impact sensitivity of four FOX-7 based PBXs can be viewed in the last paragraph of the "Initiation bond length distribution" section in the new manuscript.

Comment 3. p. 4, right column, line 6 - p.4, right column, line 8; (Results and discussion Binding energy)

What is the "hot point" theory? I know only "Hot Spot" theory.

I couldn't get reference 24 (J. Xiao, L. Zhao, W. Zhu, J. Chen, G. Ji, F. Zhao, Q. Wu and H. Xiao, Science China Chemistry, 2012, 55, 2587-2594.). So, I didn't know the details of the "hot point" theory. This confuses the reader, so authors should elaborate on it.

Sorry, my typo in the article misled you. The theory I use is the "Hot Spot" theory. This error has been corrected in the new manuscript and can be viewed in the second paragraph of the "Binding Energy" chapter. References you did not find are provided here: Molecular dynamics study on the relationships of modeling, structural and energy properties with sensitivity for RDX-based PBXs. (DOI: 10.1007/s11426-012-4797-1).

Comment 4. p. 5, right column, line 3 - p.5, right column, bottom; (Results and discussion Initiation bond length distribution)

The authors should further investigate the phenomenon that the bond length of C-NO₂ in FOX-7 is increased by the addition of four polymers from the viewpoint of intermolecular interaction. Specifically, the intermolecular distance between C-NO₂ in FOX-7 and the four polymers should be shown to the reader. In addition, the differences in intermolecular interactions between the four polymers and FOX-7 should be discussed in detail. Readers are very dissatisfied with this point.

Your suggestions are very useful. Regarding the interaction between FOX-7 and the four polymers, I added a new chapter "Radial Distribution Function" to the new manuscript. This chapter analyzes in detail the interaction between FOX-7 and four polymers. You can view it on the ninth page of the article. Thank you again for your suggestions and hope to learn more from you.

Thanks again to all editors and reviewers. It would be an honor for me to publish my article in royal society open science.

Sincerely yours,

Baoguo Wang

Appendix B

Dear Editor and Reviewer :

Thank you for your email and the reviewer's comments concerning our manuscript entitled "Computational analysis the relationships of energy and mechanical properties with sensitivity for FOX-7 based PBXs via MD simulation" (ID: RSOS-200345.R1). We thank the reviewer for the constructive comments. We revised the manuscript according to the recommendations. We provide below with a brief "List of changes" in the revised manuscript, followed by our answers to the referee's comments.

List of changes:

1. We calculated the distance between the O atom in C-NO₂ and the H atom in the four polymers, and analyzed the influence of the addition of the polymer on the C-NO₂ bond length. The revised content can be viewed in the fifth paragraph of the chapter "Initiation bond length distribution" in the new manuscript (P. 7, right column, line 3-P. 8, left column, line 9). The statistical results can be viewed in Tables 5.
2. We modify $g_{inter}(r)$ to $g(r)$. The revised content can be viewed in the first paragraph of the "Radial Distribution Function" chapter in the new manuscript (P. 9, left column, line 5).
3. We revised the hydrogen bond range to "1.1-3.1Å", and re-marked and modified the radial distribution function graph. The revised content can be viewed in the second paragraph of the "Radial Distribution Function" chapter in the new manuscript (P. 9, left column, line 18-19). The radial distribution function plots can be viewed in Figure 17 in the new manuscript and Figs S10-12 in the supporting information.
4. We have rewritten the analysis of the radial functional distribution, discussed in detail

the differences in the interactions between the four binder molecules and the FOX-7 molecule, and presented this section to the reader in as accessible a manner as possible. The revised content can be viewed in the third, fourth, and fifth paragraphs of the "Radial Distribution Function" chapter in the new manuscript (P. 9, right column, top-P. 10, left column, bottom).

Detailed reply to reviewer's comments:

Comment 1: First of all, what does "The radial distribution function $g_{\text{inter}}(r)$ " mean? I only know "The radial distribution function $g(r)$ ". Please elaborate on the meaning of "ginter" to the reader.

Response to comment 1: Sorry for any misunderstanding caused by the writing error here. Since we are calculating the radial distribution functions between molecules rather than within molecules, we want to use $g_{\text{inter}}(r)$ to represent the intermolecular radial distribution function. However, due to a writing error, we did not turn the "inter" after g into a subscript, so we apologize again for the misunderstanding. We also reviewed the literature, where the chapters on radial distribution functions are denoted by $g(r)$. So, we use $g(r)$ to represent the radial distribution function directly in the new manuscript. The completed revision can be found in the first paragraph of the "Radial Distribution Functions" section of the new manuscript (P. 9, left column, line 5). The references are as follows: ① Molecular Dynamics Simulation on TKX-50 Based Explosives. Open Access Journal of Chemistry. Volume 3, Issue 1, 2019, PP 20-24. ② Theoretical insight into the cocrystal explosive of 2,4,6,8,10,12-hexanitrohexaazaisowurtzitane (CL-20)/1-Methyl-4,5-dinitro-

1H-imidazole (MDNI). Journal of Theoretical and Computational Chemistry. Vol. 16, No. 7 (2017) 1750061 (19 pages). DOI: 10.1142/S0219633617500614.

Comment 2: What do the symbols a, b, c in Figure 16 mean? Corresponding to the text of the authors, it is thought that a represents hydrogen bond and b represents van der Waals force. Then, what does c correspond to in Figure S10, 11, and 12? ..Authors should kindly explain to the reader.

Response to comment 2: Your suggestions are very helpful and this part of the presentation does confuse the reader. We have explained the meaning and role of a, b, c, and d in the new manuscript. The completed revision can be found in the third paragraph of the "Radial Distribution Functions" chapter in the new manuscript (P. 9, right column, line 6-P. 10, left column, line 3).

Comment 3: In the description in the text of Fig. 16, "at low temperatures (198-298K), hydrogen bonds exist between H1-O2 molecules, but as the temperature increases, the hydrogen bonds gradually disappear." However, can Figure 16 really reasonably lead to such a conclusion? .. I'm curious about the new peaks that are appearing at close range. Similarly, there are explanations in various places that I cannot understand. The text of this session will only confuse the reader. The authors should explain in an easy-to-read manner to the reader.

Response to comment 3: We apologize for the lack of clarity in this section, which was caused by an oversight in our work. Due to the complexity of the data in this chapter, and our negligence in processing the data, the RDF plots in the old manuscript have

typographical errors. We also have an erroneous understanding of the range of action of hydrogen bonds. After reviewing some references, we corrected the range of hydrogen bonding action to 1.1-3.1Å. "The new peaks that are appearing at close range" are due to the presence of hydrogen bonding. In the new manuscript, we have re-labeled the peaks present in the hydrogen bonding range of action and corrected typographical errors present in the RDF plots. The corrected RDF plots can be viewed in Fig 17 in the new manuscript and Figs S10-12 in the supporting information. The analysis of the radial distribution function has been rewritten, and this section has been explained to the reader in a way that is as easy to understand as possible. The revised content can be viewed in the third, fourth, and fifth paragraphs of the "Radial Distribution Function" chapter in the new manuscript (P. 9, right column, top-P. 10, left column, bottom). References on the scope of hydrogen bonding are as follows: ① Molecular dynamic simulations on TKX-50/RDX cocrystal. Journal of Molecular Graphics and Modelling. DOI: 10.1016/j.jmglm.2017.03.006 ② Molecular dynamics simulations on dihydroxylammonium 5,5'-bistetrazole-1,1'-diolate/hexanitrohexaazaisowurtzitane cocrystal. RSC Advances. DOI: 10.1039/C5RA24924D.

Comment 4: In comment 4, I commented that it is necessary to investigate the phenomenon that the bond length of C-NO₂ in FOX-7 is decreased by the addition of four polymers from the viewpoint of intermolecular interaction. Did you know anything about this with the radial distribution function? Authors should explain to readers in detail.

In addition, the differences in intermolecular interactions between the four polymers and FOX-7 need to be elaborated.

Response to comment 4: We're sorry that we didn't fully understand what you meant in the last revision. In the new manuscript we have counted the distances between C-NO₂ and the four polymers, as you suggested, and analyzed the effect of polymer incorporation on the C-NO₂ bond length. The changes can be found in the fifth paragraph of the section "Initiation bond length distribution" in the new manuscript (P. 7, right column, line 3-P. 8, left column, line 9). The statistical results are available in Tables 5. The differences in the intermolecular interactions between the four polymers and FOX-7 are discussed in detail in the section "Radial Distribution Function". Normally, RDF can reveal the intermolecular forces¹. The revised content can be found in the third, fourth, and fifth paragraphs of the "Radial Distribution Functions" chapter in the new manuscript (P. 9, right column, top-P. 10, left column, bottom).

Reference

1. C. Wu, S. Zhang, F. Ren, R. Gou and G. Han, *Journal of Theoretical and Computational Chemistry*, 2017, **16**.

We thank the reviewer and editor, and remain at your disposal for any further questions.

Yours sincerely

Baoguo Wang

Appendix C

Dear Editor and Reviewer :

Thank you for your email and the reviewer's comments concerning our manuscript entitled "Computational analysis the relationships of energy and mechanical properties with sensitivity for FOX-7 based PBXs via MD simulation" (ID: RSOS-200345.R2). We thank the reviewer for the constructive comments. We revised the manuscript according to the recommendations. We provide below with a brief "List of changes" in the revised manuscript, followed by our answers to the referee's comments.

List of changes:

1. The title of the article has been changed from "Computational analysis the relationships of energy and mechanical properties with sensitivity for FOX-7 based PBXs via MD simulation" to "Computational analysis the relationships of energy and mechanical properties with sensitivity for 1, 1-diamino-2, 2-dinitroethene based polymer bonded explosives".
2. Added calculations for quantum chemistry. The revised content can be viewed in the first paragraph of the chapter "Quantum Chemical Calculations" in the new manuscript (P. 4, left column, line 30-P. 4, right column, line 6).
3. A section on structure analysis has been added to provide more accurate calculations of bond length parameters as well as intermolecular distance parameters. The revised content can be viewed in the first paragraph of the chapter "Structure analysis" in the new manuscript (P. 7, right column, line 22-P. 8, left column, line 19).
4. RDG analysis chapter has been added. Analysis of intermolecular interactions is more

visual and simple with RDG. The revised content can be viewed in the first paragraph and second paragraph of the chapter "RDG analysis" in the new manuscript (P. 8, right column, line 4-P. 9, left column, line 8).

Detailed reply to reviewer's comments:

To reviewer2:

Comment 1: The authors described that the hydrogen bonding and van der Waals forces of P1 are the highest compared to other systems. However, objectively, P1 and P2 are almost the same. The Lave value for P1 is minimal, but the "significant" Lmax is the second height. The authors described that Lmax is a more critical parameter than Lave.

In general, even in advanced quantum chemical calculations, discussion of the bond distance with a difference of 3 ~ 4 digits or less after the decimal point causes a problem of calculation accuracy.

Therefore, it is not convincing to the reader that molecular dynamics calculation, which is not quantum chemical calculation, advances the discussion with a difference of 3 ~ 4 decimal places or less in the bond distance.

In addition, the authors presume that the large interaction between ESTANE5703 and EVA is due to the high number of 140-180° hydrogen bonds. Generally, the strength of hydrogen bonds depends on the distance between the bonding atoms and the distribution of charges. The bond angle is only a secondary effect. The authors should show and explain the data to the reader.

Response to comment 1: Your comments are good, and the accuracy of molecular dynamics calculations is really not good enough. In order to obtain more accurate calculations, we combined FOX-7 with four polymers into four composite structures, respectively, and analyzed them computationally at the B3LYP/6-311++G (d, p) level using quantum chemical methods. The bond lengths of FOX-7 molecules were calculated with greater precision by quantum chemistry, and the changes in the bond lengths of FOX-7 molecules after the addition of polymeric molecules are listed in Table 5 of the new manuscript. We also list the distance parameters between the FOX-7 molecule and the polymeric molecule, which reflect the intermolecular hydrogen bond formation, as shown in Table 6 in the new manuscript. However, we still retained the part of the molecular dynamics calculations of the initiation bond length distribution, which is not very accurate, but its calculation system is large enough to provide statistically significant variation patterns, especially with temperature. The revised content can be viewed in the first paragraph of the chapter "Structure analysis" in the new manuscript (P. 7, right column, line 22-P. 8, left column, line 19).

Comment 2: As already mentioned in Comment 5, it is not convincing to the reader that molecular dynamics calculation, which is not quantum chemical calculation, advances the discussion due to the difference of the bonding distance to 3 ~ 4 decimal places or less. Therefore, this session will need to be significantly rewritten for the reader to understand.

Response to comment 2: As mentioned in Response to comment 1, we have performed precise calculations of bond lengths using quantum chemical methods, which can be found

in the Structural Analysis section of the new manuscript. (P. 7, right column, line 22-P. 8, left column, line 19)

Comment 3: First of all, Fig. 17 seems to be P1, but I don't know which P2 to P4 corresponds to Fig. S10-12. This point is very unfriendly to the reader. Please explain the figure so that the reader can understand it. The sentences in the red part are very difficult for the reader to understand. The authors describe - "Hydrogen bonding mainly exists in O-H atom pairs and F-H atom pairs, and there is almost no hydrogen bonding in N-H atom pairs." However, in Figure 17(B), hydrogen bonds are present in N1-H2. Further, only H1-O2 (A), N1-H2 (B) and O1-H2 (C) are shown in Fig. 17 and Fig. S10-12. The reader cannot judge the hydrogen bond of the F-H atom pairs. The authors describe - "Through comparative analysis, it is found that the interaction between FOX-7 and PEG is stronger than other systems, which is reflected in that O1-H2 and N1-H2 have the highest peak values of hydrogen bonds and strong van der Waals forces in the four systems at different temperatures." However, "is there almost no hydrogen bonding in N-H atom pairs?" From Fig. 17 and Fig. S10-12, I can not understand that the interaction between FOX-7 and PEG is clearly stronger than in other systems. The authors should improve the presentation of the figure to show the reader that the interaction between FOX-7 and PEG is stronger than other systems. The authors need to rewrite the radial distribution function diagram significantly to make it easier for the reader.

Response to comment 3: We are sorry that we have made two changes to the section on the radial distribution function, but it still does not meet your requirements. We tried other methods to analyze the intermolecular interactions of FOX-7 with polymers more simply

and clearly. The RDG analysis of the composite structure obtained from the optimization calculations on the level of B3LYP/6-311++G (d, p) was performed, and four scatter plots and iso-plots were plotted to show the interaction between FOX-7 and polymer molecules more clearly and visually. The revised content can be viewed in the first paragraph and second paragraph of the chapter "RDG analysis" in the new manuscript (P. 8, right column, line 4-P. 9, left column, line 8).

To reviewer3:

Comment 1: There should be no acronyms in the title. MD, FOX-7 and PBXs should be spelled out. The title should be shorter. I suggest something like "Exploring shock sensitivity of high energy materials with molecular dynamics".

Response to comment 1: Thank you for your suggestion. The main purpose of this paper is to investigate how the binding energy, cohesive energy density, initiation bond length distribution, and mechanical properties of PBXs change as the temperature increases and the explosive sensitivity increases. Based on these changes, we investigate which properties can be used as the basis for determining the sensitivity of FOX-7-based PBX explosives. Therefore, the article mainly wants to discuss the relationship between some microscopic properties and sensitivity of energetic materials, and the screening of PBXs formulations, rather than purely studying the sensitivity. Based on your suggestions, we have spelled the acronym with the full letter. Although the content does not refer to your suggestions, thank you for the suggested changes to the title of the article.

Comment 2: In the construction of polymer models, please explain the "Smart" method

the "Anderson" thermostat and the "Fine" quality for those readers unfamiliar with COMPASS and whatever other software the authors are using.

Response to comment 2: Your suggestion is good. We have made changes based on your suggestion. The method is described in S.4 in the support information.

Comment 3: A mention of the mechanism by which FOX-7 explodes would be very valuable.

I am confused, in particular, by the very elevated high-temperature simulations. I thought polynitrated compounds explode via radical branching not via thermochemical acceleration. In fact it may be the case that at different Ts one mechanism prevails over the other.

Response to comment 3: Thank you for your suggestion. It may be that I was unclear in my original statement, and you seem to have misunderstood the purpose of our calculations at high temperatures. You are right that the mechanism you mentioned is indeed more important when studying the explosion mechanism. But the purpose of this paper is not to study the explosion mechanism of FOX-7. As I mentioned in Response to comment 1, what we want to study is a relationship between the microscopic properties of PBXs and its sensitivity. The simulations at high temperatures were performed to verify whether the microscopic properties calculated at low temperatures have a similar pattern of variation at high temperatures. The purpose of this is to further establish that the microscopic properties we are examining are indeed correlated with the sensitivity of the explosive. The part about misleading you has been revised in the new manuscript (P. 3,

right column, line 21-25). The decomposition and explosion mechanism of FOX-7 has been reported, see DOI: 10.1063/1.4896165

Comment 4: In the initiation bond length distribution: Figure 11 is too busy. One bond length distribution is probably sufficient to make the point.

Response to comment 4: Thank you for your careful reading of the article and your suggestions for revisions. It is true that Figure 11 is too busy, but Figure 11 can be clearly contrasted with Figure 14, which allows the reader to easily observe the significant difference in the bond length distribution of the initiator after low and high temperature calculations. Therefore, after consideration, we have kept Figure 11 intact.

Comment 5: In the same section, how is L_{\max} calculated?

Response to comment 5: After counting all the C-NO₂ bonds in the system, the maximum bond length L_{\max} is obtained.

Comment 6: In the conclusion section the authors talk about the binding energy.

This needs to be clarified. The binding energies of what? Relative to what?

Response to comment 6: The binding energy in the article mainly refers to the interaction energy between FOX-7 molecules and polymer molecules, which is related to the thermal stability of the PBXs system.

Comment 7: Finally, and most importantly, the authors need to do a better job at convincing the reader

that the mechanical and structural properties they compute do in fact explain

how certain polymer blends do a better job at decreasing the sensitivity of high energy materials. Is there evidence in the literature that the authors approach actually works?

How do their results compare with experiment? To me, without a thorough understanding of the mechanism and a better characterization of the force fields with some barriers (transition states) is not sufficient evidence that the various materials considered actually do decrease the sensitivity of FOX-7.

Response to comment 7: Your advice is excellent. There is much more literature similar to our work. The addition of polymers is effective in reducing the sensitivity of explosives, as has been demonstrated in many works. Some relevant literature is provided here for your reference. ① Molecular dynamic simulations for FOX-7 and FOX-7 based PBXs (10.1007/s00894-018-3687-7). ② Molecular dynamics simulations on 3-CL-20-based PBXs with added GAP and its derivative polymers (10.1039/c7ra13517c). ③ Molecular dynamics study on the relationships of modeling, structural and energy properties with sensitivity for RDX-based PBXs (10.1007/s11426-012-4797-1). ④ Investigation into the Coating and Desensitization Effect on HNIW of Paraffin Wax/Stearic Acid Composite System (10.1080/07370652.2014.993049).

We thank the reviewer and editor, and remain at your disposal for any further questions.

Yours sincerely

Baoguo Wang